# LOGIC AND THE 2-SIMPLICIAL TRANSFORMER

**James Clift** [*]      **Dmitry Doryn** [*]      **Daniel Murfet** [†]      **James Wallbridge** [*]

[*] {jamesedwardclift,dmitry.doryn,james.wallbridge}@gmail.com
[†] Department of Mathematics, University of Melbourne, d.murfet@unimelb.edu.au

## ABSTRACT

We introduce the 2-simplicial Transformer, an extension of the Transformer which includes a form of higher-dimensional attention generalising the dot-product attention, and uses this attention to update entity representations with tensor products of value vectors. We show that this architecture is a useful inductive bias for logical reasoning in the context of deep reinforcement learning.

## 1 INTRODUCTION

Deep learning contains many differentiable algorithms for computing with learned representations. These representations form vector spaces, sometimes equipped with additional structure. A recent example is the Transformer (Vaswani et al., 2017) in which there is a vector space $V$ of *value vectors* and an inner product space $H$ of *query and key vectors*. This structure supports a kind of message-passing, where a value vector $v_j \in V$ derived from entity $j$ is propagated to update an entity $i$ with weight $q_i \cdot k_j$, where $q_i \in H$ is a query vector derived from entity $i$, $k_j \in H$ is a key vector derived from entity $j$, and the inner product on $H$ is written as a dot product.

The Transformer therefore represents a *relational* inductive bias, where a relation from entity $j$ to entity $i$ is perceived to the extent that $q_i \cdot k_j$ is large and positive. However, the real world has structure beyond entities and their direct relationships: for example, the three blocks in Figure 1 are arranged in such a way that if either of the supporting blocks is removed, the top block will fall. This is a simple 3-way relationship between entities $i, j, k$ that is complex to represent as a system of 2-way relationships. It is natural to make the hypothesis that such higher-order relationships are essential to extracting the full predictive power of data, across many domains.

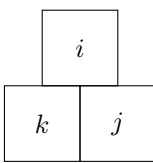

Figure 1: A 3-way relationship between blocks.

In accordance with this hypothesis, we introduce a generalisation of the Transformer architecture, the 2-*simplicial Transformer*, which incorporates both 2- and 3-way interactions. Mathematically, the key observation is that higher-order interactions between entities can be understood using *algebras*. This is nothing but Boole's insight (Boole, 1847) which set in motion the development of modern logic. In our situation, an appropriate algebra is the Clifford algebra $\mathrm{Cl}(H)$ of the space $H$ of queries and keys, which contains that space $H \subseteq \mathrm{Cl}(H)$ and in which queries and keys can be multiplied.

To represent a 3-way interaction we map each entity $i$ to a triple $(p_i, l_i^1, l_i^2)$ of vectors in $H$ consisting of a query vector $p_i$, a (first) key vector $l_i^1$ and a (second) key vector $l_i^2$. Given a triple $i, j, k$ we first form the product $p_i l_j^1 l_k^2$ in the Clifford algebra, and then extract a scalar quantity $\eta(p_i l_j^1 l_k^2)$ using a natural continuous function $\eta : \mathrm{Cl}(H) \longrightarrow \mathbb{R}$ associated to the $\mathbb{Z}$-grading of $\mathrm{Cl}(H)$. This scalar

---

[*]Listing order is alphabetical. Correspondence to d.murfet@unimelb.edu.au.

measures how strongly the network perceives a 3-way interaction involving $i, j, k$. In summary, the 2-simplicial Transformer learns how to represent entities in its environment as vectors $v \in V$, and how to transform those entities to queries and (pairs of) keys in $H$, so that the signals provided by the scalars $q_i \cdot k_j$ and $\eta(p_i l_j^1 l_k^2)$ are informative about higher-order structure in the environment.

As a toy example of higher-order structure, we consider the reinforcement learning problem in a variant of the BoxWorld environment from (Zambaldi et al., 2019). The original BoxWorld is played on a rectangular grid populated by keys and locked boxes of varying colours, with the goal being to open the box containing the "Gem". In our variant of the BoxWorld environment, *bridge BoxWorld*, the agent must use two keys simultaneously to obtain the Gem; this structure in the environment creates many 3-way relationships between entities, including for example the relationship between the locked boxes $j, k$ providing the two keys and the Gem entity $i$. This structure in the environment is fundamentally *logical* in nature, and encodes a particular kind of conjunction; see Appendix I.

The architecture of our deep reinforcement learning agent largely follows (Zambaldi et al., 2019) and the details are given in Section 4. The key difference between our *simplicial agent* and the *relational agent* of (Zambaldi et al., 2019) is that in place of a standard Transformer block we use a 2-simplicial Transformer block. Our experiments show that the simplicial agent confers an advantage over the relational agent as an inductive bias in our reasoning task. Motivation from neuroscience for a simplicial inductive bias for abstract reasoning is contained in Appendix J.

Our use of tensor products of value vectors is inspired by the semantics of linear logic in vector spaces (Girard, 1987; Melliès, 2009; Clift & Murfet, 2017; Wallbridge, 2018) in which an algorithm with multiple inputs computes on the tensor product of those inputs, but this is an old idea in natural language processing, used in models including the second-order RNN (Giles et al., 1989; Pollack, 1991; Goudreau et al., 1994; Giles et al., 1991), multiplicative RNN (Sutskever et al., 2011; Irsoy & Cardie, 2015), Neural Tensor Network (Socher et al., 2013) and the factored 3-way Restricted Boltzmann Machine (Ranzato et al., 2010), see Appendix A. Tensors have been used to model predicates in a number of neural network architectures aimed at logical reasoning (Serafini & Garcez, 2016; Dong et al., 2019). The main novelty in our model lies in the introduction of the 2-simplicial attention, which allows these ideas to be incorporated into the Transformer architecture.

## 2 2-SIMPLICIAL TRANSFORMER

In this section we first review the definition of the ordinary Transformer block and then explain the 2-simplicial Transformer block. We distinguish between the *Transformer architecture* which contains a word embedding layer, an encoder and a decoder (Vaswani et al., 2017), and the *Transformer block* which is the sub-model of the encoder that is repeated. The fundamental idea, of propagating information between nodes using weights that depend on the dot product of vectors associated to those nodes, comes ultimately from statistical mechanics via the Hopfield network (Appendix B).

The ordinary and 2-simplicial Transformer blocks define operators on sequences $e_1, \ldots, e_N$ of *entity representations*. Strictly speaking the entities are indices $1 \le i \le N$ but we sometimes identify the entity $i$ with its representation $e_i$. The space of entity representations is denoted $V$, while the space of query, key and value vectors is denoted $H$. We use only the vector space structure on $V$, but $H = \mathbb{R}^d$ is an inner product space with the usual dot product pairing $(h, h') \mapsto h \cdot h'$ and in defining the 2-simplicial Transformer block we will use additional algebraic structure on $H$, including the "multiplication" tensor $B : H \otimes H \longrightarrow H$ of (10) (used to propagate tensor products of value vectors) and the Clifford algebra of $H$ (used to define the 2-simplicial attention).

In the first step of the standard Transformer block we generate from each entity $e_i$ a tuple of vectors via a learned linear transformation $E : V \longrightarrow H^{\oplus 3}$. These vectors are referred to respectively as *query*, *key* and *value* vectors and we write

$$(q_i, k_i, v_i) = E(e_i). \tag{1}$$

Stated differently, $q_i = W^Q e_i, k_i = W^K e_i, v_i = W^V e_i$ for weight matrices $W^Q, W^K, W^V$. In the second step we compute a refined value vector for each entity

$$v_i' = \sum_{j=1}^{N} \frac{e^{q_i \cdot k_j}}{\sum_{s=1}^{N} e^{q_i \cdot k_s}} v_j = \sum_{j=1}^{N} \text{softmax}(q_i \cdot k_1, \ldots, q_i \cdot k_N)_j v_j. \tag{2}$$

Finally, the new entity representation $e_i'$ is computed by the application of a feedforward network $g_\theta$, layer normalisation and a skip connection

$$e_i' = \text{LayerNorm}\left(g_\theta(v_i') + e_i\right). \tag{3}$$

**Remark 2.1.** In the introduction we referred to the idea that a Transformer model learns *representations of relations*. To be more precise, these representations are *heads*, each of which determines an independent set of transformations $W^Q, W^K, W^V$ which extract queries, keys and values from entities. Thus a head determines not only which entities are related (via $W^Q, W^K$) but also what information to transmit between them (via $W^V$). In *multiple-head* attention with $K$ heads, there are $K$ channels along which to propagate information between every pair of entities, each of dimension $\dim(H)/K$. More precisely, we choose a decomposition $H = H_1 \oplus \cdots \oplus H_K$ so that

$$E : V \longrightarrow \bigoplus_{u=1}^{K}(H_u^{\oplus 3})$$

and write

$$(q_{i,(1)}, k_{i,(1)}, v_{i,(1)}, \ldots, q_{i,(K)}, k_{i,(K)}, v_{i,(K)}) = E(e_i).$$

To compute the output of the attention, we take a direct sum of the value vectors propagated along every one of these $K$ channels, as in the formula

$$e_i' = \text{LayerNorm}\left(g_\theta\Big[\bigoplus_{u=1}^{K}\sum_{j=1}^{N}\text{softmax}(q_{i,(u)} \cdot k_{1,(u)}, \ldots, q_{i,(u)} \cdot k_{N,(u)})_j v_{j,(u)}\Big] + e_i\right). \tag{4}$$

In combinatorial topology the canonical one-dimensional object is the 1-simplex (or edge) $j \longrightarrow i$. Since the standard Transformer model learns *representations of relations*, we refer to this form of attention as 1-*simplicial attention*. The canonical two-dimensional object is the 2-simplex (or triangle) which we may represent diagrammatically in terms of indices $i, j, k$ as

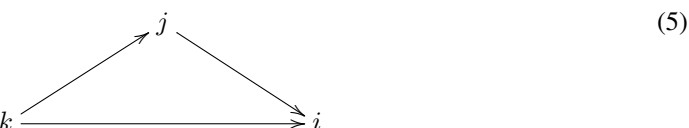

$$(5)$$

In the 2-simplicial Transformer block, in addition to the 1-simplicial contribution, each entity $e_i$ is updated as a function of pairs of entities $e_j, e_k$ using the tensor product of value vectors $u_j \otimes u_k$ and a probability distribution derived from a scalar triple product $\langle p_i, l_j^1, l_k^2 \rangle$ in place of the scalar product $q_i \cdot k_j$. This means that we associate to each entity $e_i$ a four-tuple of vectors via a learned linear transformation $E : V \longrightarrow H^{\oplus 4}$, denoted

$$(p_i, l_i^1, l_i^2, u_i) = E(e_i). \tag{6}$$

We still refer to $p_i$ as the *query*, $l_i^1, l_i^2$ as the *keys* and $u_i$ as the *value*. Stated differently, $p_i = W^P e_i, l_i^1 = W^{L_1} e_i, l_i^2 = W^{L_2} e_i$ and $u_i = W^U e_i$ for weight matrices $W^P, W^{L_1}, W^{L_2}, W^U$.

**Definition 2.2.** The *unsigned scalar triple product* of $a, b, c \in H$ is

$$\langle a, b, c \rangle = \big\|(a \cdot b)c - (a \cdot c)b + (b \cdot c)a\big\| \tag{7}$$

whose square is a polynomial in the pairwise dot products

$$\langle a, b, c \rangle^2 = (a \cdot b)^2(c \cdot c) + (b \cdot c)^2(a \cdot a) + (a \cdot c)^2(b \cdot b) - 2(a \cdot b)(a \cdot c)(b \cdot c). \tag{8}$$

This scalar triple product has a simple geometric interpretation in terms of the volume of the tetrahedron with vertices $0, a, b, c$. To explain, recall that the triangle spanned by two unit vectors $a, b$ in $\mathbb{R}^2$ has an area $A$ which can be written in terms of the dot product of $a$ and $b$. In three dimensions, the analogous formula involves the volume $V$ of the tetrahedron with vertices given by unit vectors $a, b, c$, and the scalar triple product as shown in Figure 2.

In general, given nonzero vectors $a, b, c$ let $\hat{a}, \hat{b}, \hat{c}$ denote unit vectors in the same directions. Then we can by Lemma C.10(v) factor out the length in the scalar triple product

$$\langle a, b, c \rangle = \|a\|\|b\|\|c\|\langle \hat{a}, \hat{b}, \hat{c} \rangle \tag{9}$$

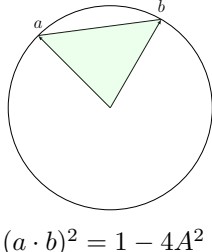 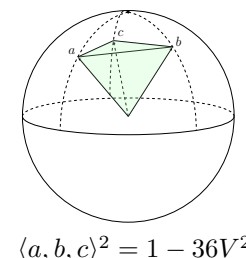

$$(a \cdot b)^2 = 1 - 4A^2 \qquad\qquad \langle a, b, c \rangle^2 = 1 - 36V^2$$

Figure 2: The geometry of 1- and 2-simplicial attention. *Left:* the dot product in terms of the area $A$ in $\mathbb{R}^2$. *Right:* the triple product in terms of the volume $V$ in $\mathbb{R}^3$.

so that a general scalar triple product can be understood in terms of the vector norms and configurations of three points on the 2-sphere. One standard approach to calculating volumes of such tetrahedrons is the cross product which is only defined in three dimensions. Since the space of representations $H$ is high dimensional the natural framework for the triple scalar product $\langle a, b, c \rangle$ is instead the Clifford algebra of $H$ (see Appendix C).

For present purposes, we need to know that $\langle a, b, c \rangle$ attains its minimum value (which is zero) when $a, b, c$ are pairwise orthogonal, and attains its maximum value (which is $\|a\|\|b\|\|c\|$) if and only if $\{a, b, c\}$ is linearly dependent (Lemma C.10). Using the number $\langle p_i, l_j^1, l_k^2 \rangle$ as a measure of the degree to which entity $i$ is attending to $(j, k)$, or put differently, the degree to which the network predicts the existence of a 2-simplex $(i, j, k)$, the update rule for the entities when using purely 2-simplicial attention is

$$v'_i = \sum_{j,k=1}^N \frac{e^{\langle p_i, l_j^1, l_k^2 \rangle}}{\sum_{s,t=1}^N e^{\langle p_i, l_s^1, l_t^2 \rangle}} B(u_j \otimes u_k) \tag{10}$$

where $B : H \otimes H \longrightarrow H$ is a learned linear transformation. Although we do not impose any further constraints, the motivation here is to equip $H$ with the structure of an algebra; in this respect we model conjunction by multiplication, an idea going back to Boole (Boole, 1847).

We compute multiple-head 2-simplicial attention in the same way as in the 1-simplicial case. To combine 1-simplicial heads (that is, ordinary Transformer heads) and 2-simplicial heads we use separate inner product spaces $H^1, H^2$ for each simplicial dimension, so that there are learned linear transformations $E^1 : V \longrightarrow (H^1)^{\oplus 3}, E^2 : V \longrightarrow (H^2)^{\oplus 4}$ and the queries, keys and values are extracted from an entity $e_i$ according to

$$(q_i, k_i, v_i) = E^1(e_i) \,,$$
$$(p_i, l_i^1, l_i^2, u_i) = E^2(e_i) \,.$$

The update rule (for a single head in each simplicial dimension) is then:

$$v'_i = \Big\{ \sum_{j=1}^N \frac{e^{q_i \cdot k_j}}{\sum_{s=1}^N e^{q_i \cdot k_s}} v_j \Big\} \oplus \mathrm{LayerNorm} \Big\{ \sum_{j,k=1}^N \frac{e^{\langle p_i, l_j^1, l_k^2 \rangle}}{\sum_{s,t=1}^N e^{\langle p_i, l_s^1, l_t^2 \rangle}} B(u_j \otimes u_k) \Big\}, \tag{11}$$

$$e'_i = \mathrm{LayerNorm}\big(g_\theta(v'_i) + e_i\big). \tag{12}$$

If there are $K_1$ heads of 1-simplicial attention and $K_2$ heads of 2-simplicial attention, then (11) is modified in the obvious way using $H^1 = \bigoplus_{u=1}^{K_1} H_u^1$ and $H^2 = \bigoplus_{u=1}^{K_2} H_u^2$.

**Remark 2.3.** Without the additional layer normalisation on the output of the 2-simplicial attention we find that training is unstable. The natural explanation is that these outputs are constructed from polynomials of higher degree than the 1-simplicial attention, and thus computational paths that go through the 2-simplicial attention will be more vulnerable to exploding or vanishing gradients.

The time complexity of 1-simplicial attention as a function of the number of entities is $O(N^2)$ while the time complexity of 2-simplicial attention is $O(N^3)$ since we have to calculate the attention for every triple $(i, j, k)$ of entities. For this reason we consider only triples $(i, j, k)$ where the base of the 2-simplex $(j, k)$ is taken from a set of pairs predicted by the ordinary attention, which we view as

the primary locus of computation. More precisely, we introduce in addition to the $N$ entities (now referred to as *standard* entities) a set of $M$ *virtual* entities $e_{N+1}, \ldots, e_{N+M}$. These virtual entities serve as a "scratch pad" onto which the iterated ordinary attention can write representations, and we restrict $j, k$ to lie in the range $N < j, k \le N + M$ so that only value vectors obtained from virtual entities are propagated by the 2-simplicial attention.

With virtual entities the update rule is for $1 \le i \le N$

$$v_i' = \left\{ \sum_{j=1}^{N} \frac{e^{q_i \cdot k_j}}{\sum_{s=1}^{N} e^{q_i \cdot k_s}} v_j \right\} \oplus \mathrm{LayerNorm} \left\{ \sum_{j,k=N+1}^{N+M} \frac{e^{\langle p_i, l_j^1, l_k^2 \rangle}}{\sum_{s,t=1}^{N+M} e^{\langle p_i, l_l^1, l_m^2 \rangle}} B(u_j \otimes u_k) \right\} \quad (13)$$

and for $N < i \le N + M$

$$v_i' = \left\{ \sum_{j=1}^{N+M} \frac{e^{q_i \cdot k_j}}{\sum_{s=1}^{N+M} e^{q_i \cdot k_s}} v_j \right\} \oplus \mathrm{LayerNorm}(u_i) \,. \quad (14)$$

The updated representation $e_i'$ is computed from $v_i', e_i$ using (12) as before. Observe that the virtual entities are not used to update the standard entities during 1-simplicial attention and the 2-simplicial attention is not used to update the virtual entities; instead the second summand in (14) involves the vector $u_i = W^U e_i$, which adds recurrence to the update of the virtual entities. After the attention phase the virtual entities are discarded.

The method for updating the virtual entities is similar to the role of the memory nodes in the relational recurrent architecture of (Santoro et al., 2018), the master node in (Gilmer et al., 2017, §5.2) and memory slots in the Neural Turing Machine (Graves et al., 2014). The update rule has complexity $O(NM^2)$ and so if we take $M$ to be of order $\sqrt{N}$ we get the desired complexity $O(N^2)$.

## 3 RL ENVIRONMENT

The environment in our reinforcement learning problem is a variant of the BoxWorld environment from (Zambaldi et al., 2019). The standard BoxWorld environment is a rectangular grid in which are situated the *player* (a dark gray tile) and a number of *locked boxes* represented by a pair of horizontally adjacent tiles with a tile of colour $x$, the *key colour*, on the left and a tile of colour $y$, the *lock colour*, on the right. There is also one *loose key* in each episode, which is a coloured tile not initially adjacent to any other coloured tile. All other tiles are blank (light gray) and are traversable by the player. The rightmost column of the screen is the *inventory*, which fills from the top and contains keys that have been collected by the player. The player can pick up any loose key by walking over it. In order to open a locked box, with key and lock colours $x, y$, the player must step on the lock while in possession of a copy of $y$, in which case one copy of this key is removed from the inventory and replaced by a key of colour $x$.

The goal is to attain a white key, referred to as the *Gem* (represented by a white square) as shown in the sample episode of Figure 3. In this episode, there is a loose pink key (marked 1) which can be used to open one of two locked boxes, obtaining in this way either key 5 or key 2[1]. The correct choice is 2, since this leads via the sequence of keys 3, 4 to the Gem.

Some locked boxes, if opened, provide keys that are not useful for attaining the Gem. Since each key may only be used once, opening such boxes means the episode is rendered unsolvable. Such boxes are called *distractors*. An episode ends when the player either obtains the Gem (with a reward of +10) or opens a distractor box (reward −1). Opening any non-distractor box, or picking up a loose key, garners a reward of +1. The *solution length* is the number of locked boxes (including the one with the Gem) in the episode on the path from the loose key to the Gem.

Our variant of the BoxWorld environment, *bridge BoxWorld*, is shown in Figure 4. In each episode two keys are now required to obtain the Gem, and there are therefore two loose keys on the board. To obtain the Gem, the player must step on either of the lock tiles with both keys in the inventory, at which point the episode ends with the usual +10 reward. Graphically, Gems with multiple locks are denoted with two vertical white tiles on the left, and the two lock tiles on the right. Two solution

---

[1]The agent sees only the colours of tiles, not the numbers which are added here for exposition.

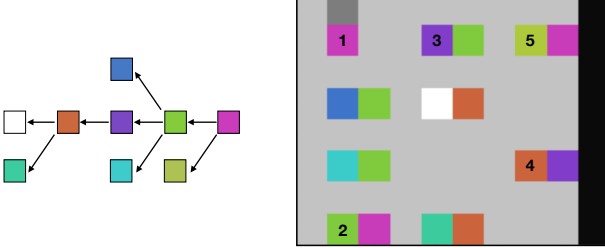

Figure 3: *Right:* a sample episode of the BoxWorld environment. The rightmost column is the player inventory, currently empty. *Left:* graph representation of the puzzle, with key colours as vertices and an arrow $C \longrightarrow D$ if key $C$ can be used to obtain key $D$.

paths (of the same length) leading to each of the locks on the Gem are generated with no overlapping colours, beginning with two loose keys. In episodes with multiple locks we do not consider distractor boxes of the old kind; instead there is a new type of distractor that we call a *bridge*. This is a locked box whose lock colour is taken from one solution branch and whose key colour is taken from the other branch. Opening the bridge renders the puzzle unsolvable. An episode ends when the player either obtains the Gem (reward $+10$) or opens the bridge (reward $-1$). Opening a box other than the bridge, or picking up a loose key, has a reward of $+1$ as before. In this paper we consider episodes with zero or one bridge (the player cannot fail to solve an episode with no bridge).

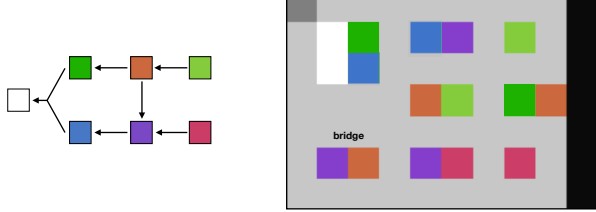

Figure 4: *Right*: a sample episode of the bridge BoxWorld environment, in which the Gem has two locks and there is a marked bridge. *Left*: graph representation of the puzzle, with upper and lower solutions paths and the bridge between them. There is a source involving the orange key and a sink involving the purple lock.

Standard BoxWorld is straightforward for an agent to solve using relational reasoning, because leaves on the solution graph can be identified (their key colour appears only once on the board) and by propagating this information backwards along the arrows on the solution graph, an agent can identify distractors. Bridge BoxWorld emphasises reasoning about 3-way relationships (or 2-simplices). The following 2-simplex motifs

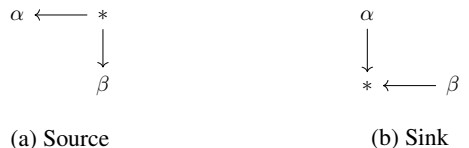

(a) Source             (b) Sink

appear in all solution graphs where a pair of boxes $(\alpha, \beta)$ is a *source* if they have the same lock colour but distinct key colours, and a *sink* if they have the same key colour but distinct lock colours (the 2-simplex leading to the Gem being an example). If $\alpha, \beta$ is a source or a sink then either $\alpha$ is the bridge or $\beta$ is the bridge. If the agent can observe both a source and a sink then it can locate the bridge. It is less clear how to identify bridges using iterated relational reasoning, because every path in the solution graph eventually reaches the Gem.

## 4 RL AGENT ARCHITECTURE

Our baseline relational agent is modeled closely on (Zambaldi et al., 2019) except that we found that a different arrangement of layer normalisations worked better in our experiments, see Remark 4.1. The code for our implementation of both agents is available online (Clift et al., 2019). In the following we describe the network architecture of both the relational and simplicial agent; we will note the differences between the two models as they arise.

The input to the agent's network is an RGB image, represented as a tensor of shape $[R, C + 1, 3]$ (i.e. an element of $\mathbb{R}^R \otimes \mathbb{R}^{C+1} \otimes \mathbb{R}^3$) where $R$ is the number of rows and $C$ the number of columns (the $C + 1$ is due to the inventory). This tensor is divided by $255$ and then passed through a $2 \times 2$ convolutional layer with 12 features, and then a $2 \times 2$ convolutional layer with 24 features. Both activation functions are ReLU and the padding on our convolutional layers is "valid" so that the output has shape $[R - 2, C - 1, 24]$. We then multiply by a weight matrix of shape $24 \times 62$ to obtain a tensor of shape $[R - 2, C - 1, 62]$. Each feature vector has concatenated to it a two-dimensional positional encoding, and then the result is reshaped into a tensor of shape $[N, 64]$ where $N = (R - 2)(C - 1)$ is the number of Transformer entities. This is the list $(e_1, \ldots, e_N)$ of entity representations $e_i \in V = \mathbb{R}^{64}$.

In the case of the simplicial agent, a further two learned embedding vectors $e_{N+1}, e_{N+2}$ are added to this list; these are the virtual entities. So with $M = 0$ in the case of the relational agent and $M = 2$ for the simplicial agent, the entity representations form a tensor of shape $[N + M, 64]$. This tensor is then passed through two iterations of the Transformer block (either purely 1-simplicial in the case of the relational agent, or including both 1 and 2-simplicial attention in the case of the simplicial agent). In the case of the simplicial agent the virtual entities are then discarded, so that in both cases we have a sequence of entities $e''_1, \ldots, e''_N$. Inside each block are two feedforward layers separated by a ReLU activation with 64 hidden nodes; the weights are shared between iterations of the Transformer block. In the 2-simplicial Transformer block the input tensor, after layer normalisation, is passed through the 2-simplicial attention and the result (after an additional layer normalisation) is concatenated to the output of the 1-simplicial attention heads before being passed through the feedforward layers. The pseudo-code for the ordinary and 2-simplicial Transformer blocks are:

```
def transformer_block(e):
    x = LayerNorm(e)
    a = 1SimplicialAttention(x)
    b = DenseLayer1(a)
    c = DenseLayer2(b)
    r = Add([e,c])
    eprime = LayerNorm(r)
    return eprime
```

```
def simplicial_transformer_block(e):
    x = LayerNorm(e)
    a1 = 1SimplicialAttention(x)
    a2 = 2SimplicialAttention(x)
    a2n = LayerNorm(a2)
    ac = Concatenate([a1,a2n])
    b = DenseLayer1(ac)
    c = DenseLayer2(b)
    r = Add([e,c])
    eprime = LayerNorm(r)
    return eprime
```

Our implementation of the standard Transformer block is based on an implementation in Keras from (Mavreshko, 2019). In both the relational and simplicial agent, the space $V$ of entity representations has dimension $64$ and we denote by $H^1, H^2$ the spaces of 1-simplicial and 2-simplicial queries, keys and values. In both the relational and simplicial agent there are two heads of 1-simplicial attention, $H^1 = H^1_1 \oplus H^1_2$ with $\dim(H^1_i) = 32$. In the simplicial agent there is a single head of 2-simplicial attention with $\dim(H^2) = 48$ and two virtual entities.

The output of the Transformer blocks is a tensor of shape $[N, 64]$. To this final entity tensor we apply max-pooling over the entity dimension, that is, we compute a vector $v \in \mathbb{R}^{64}$ by the rule $v_i = \max_{1 \le j \le N}(e''_j)_i$ for $1 \le i \le 64$. This vector $v$ is then passed through four fully-connected layers with $256$ hidden nodes and ReLU activations. The output of the final fully-connected layer is multiplied by one $256 \times 4$ weight matrix to produce logits for the actions (left, up, right and down) and another $256 \times 1$ weight matrix to produce the value function.

**Remark 4.1.** There is wide variation in the layer normalisation in Transformer models, compare (Vaswani et al., 2017; Child et al., 2019; Zambaldi et al., 2019). In (Zambaldi et al., 2019) layer normalisation occurs in two places: on the concatenation of the $Q, K, V$ matrices, and on the output of the feedforward network $g_\theta$. We keep this second normalisation but move the first from *after* the

linear transformation $E$ of (1) to *before* this linear transformation, so that it is applied directly to the incoming entity representations. This ordering gave the best performant relational model in our experiments, with our results diverging even further if a direct comparison to the (Zambaldi et al., 2019) architecture was used.

## 5 EXPERIMENTS AND RESULTS

The training of our agents uses the implementation in Ray RLlib (Liang et al., 2018) of the distributed off-policy actor-critic architecture IMPALA of (Espeholt et al., 2018) with optimisation algorithm RMSProp. The hyperparameters for IMPALA and RMSProp are given in Table 1 of Appendix E. Following (Zambaldi et al., 2019) and other recent work in deep reinforcement learning, we use RMSProp with a large value of the hyperparameter $\varepsilon = 0.1$. As we explain in Appendix G, this is effectively RMSProp with smoothed gradient clipping.

First we verified that our implementation of the relational agent solves the BoxWorld environment (Zambaldi et al., 2019) with a solution length sampled from $[1, 5]$ and number of distractors sampled from $[0, 4]$ on a $9 \times 9$ grid. After training for $2.35 \times 10^9$ timesteps our implementation solved over 93% of puzzles (regarding the discrepancy with the reported sample complexity in (Zambaldi et al., 2019) see Appendix D). Next we trained the relational and simplicial agent on bridge BoxWorld, under the following conditions: half of the episodes contain a bridge, the solution length is uniformly sampled from $[1, 3]$ (both solution paths are of the same length), colours are uniformly sampled from a set of 20 colours and the boxes and loose keys are arranged randomly on a $7 \times 9$ grid, under the constraint that the box containing the Gem does not occur in the rightmost column or bottom row, and keys appear only in positions $(y, x) = (2r, 3c - 1)$ for $1 \leq r \leq 3, 1 \leq c \leq 3$. The starting and ending point of the bridge are uniformly sampled with no restrictions (e.g. the bridge can involve the colours of the loose keys and locks on the Gem) but the lock colour is always on the top solution path. There is no curriculum and no cap on timesteps per episode.

We trained four independent trials of both agents to either $5.5 \times 10^9$ timesteps or convergence, whichever came first. In Figure 6 we give the mean and standard deviation of these four trials, showing a clear advantage of the simplicial agent. We make some remarks about performance comparisons taking into account the fact that the relational agent is simpler (and hence faster to execute) than the simplicial agent in Appendix D. The training runs for the relational and simplicial agents are shown in Figure 9 and Figure 10 of Appendix F, together with analysis and visualization of the 1- and 2-simplicial attention in specific examples.

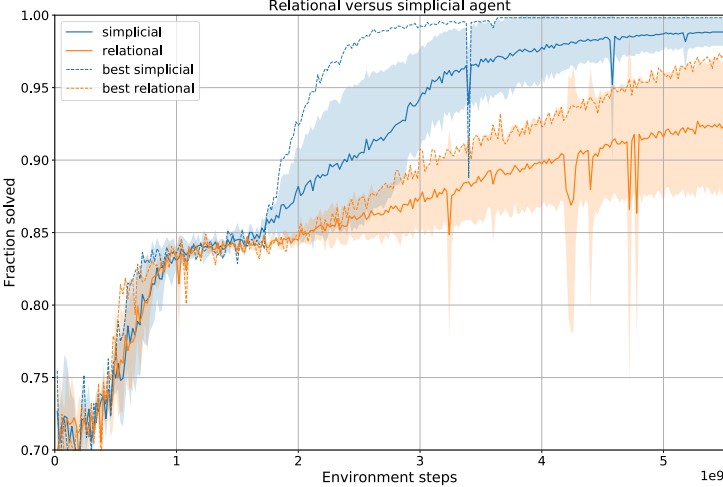

Figure 6: Training curve of mean relational and simplicial agents on bridge BoxWorld. Shown are the mean and standard deviation of four runs of each agent, including the best run of each.

In the reported experiments we use only two Transformer blocks; we performed two trials of a relational agent using four Transformer blocks, but after $5.5 \times 10^9$ timesteps neither trial exceeded

the $0.85$ plateau in terms of fraction solved. Our overall results therefore suggest that the 2-simplicial Transformer is more powerful than the standard Transformer, with its performance not matched by adding greater depth. This is further supported by the fact on a time-adjusted basis, the 2-simplicial model still converges faster than the ordinary model; see Figure 8 of Appendix D.

## 6 ANALYSIS

We analyse the simplicial agent to establish that it has learned to use the 2-simplicial attention, and to provide some intuition for why 2-simplices are useful; additional details are in Appendix F. The analysis is complicated by the fact that our $2 \times 2$ convolutional layers (of which there are two) are not padded, so the number of entities processed by the Transformer blocks is $(R-2)(C-1)$ where the original game board is $R \times C$ and there is an extra column for the inventory (here $R$ is the number of rows). This means there is not a one-to-one correspondence between game board tiles and entities; for example, all the experiments reported in Figure 6 are on a $7 \times 9$ board, so that there are $N = 40$ Transformer entities which can be arranged on a $5 \times 8$ grid (information about this grid is passed to the Transformer blocks via the positional encoding). Nonetheless we found that for trained agents there is a strong relation between a tile in position $(y, x)$ and the Transformer entity with index $x + (C-1)(y-1) - 1$ for $(y, x) \in [1, R-2] \times [1, C-1] \subseteq [0, R-1] \times [0, C]$. This correspondence is presumed in the following analysis, and in our visualisations.

Displayed in Figure 7 are attention distributions for simplicial agent A of Figure 10. The four images in the top right show the ordinary attention of the virtual entities in the first iteration of the simplicial Transformer block: in the first head, the first virtual entity attends strongly to a particular lock, while the second head of the second virtual entity attends strongly to the corresponding key. Shown at the bottom of Figure 7 is the 2-simplicial attention in the second iteration of the simplicial Transformer block. The columns are query entities $i$ and rows are key entity pairs $(j, k)$ in lexicographic order $(1, 1), (1, 2), (2, 1), (2, 2)$. Entity 17 is the top lock on the Gem, 25 is the bottom lock on the Gem, 39 is the player. We may therefore infer, from our earlier description of the ordinary attention of the virtual entities, that the agent "perceives" the 2-simplex with query entity 25 as shown. In general we observe that the top and bottom locks on the Gem, the player, and the entities $7, 15$ associated to the inventory often have a non-generic 2-simplicial attention, which strongly suggests that the simplicial agent has learned to use 2-simplices in a meaningful way.

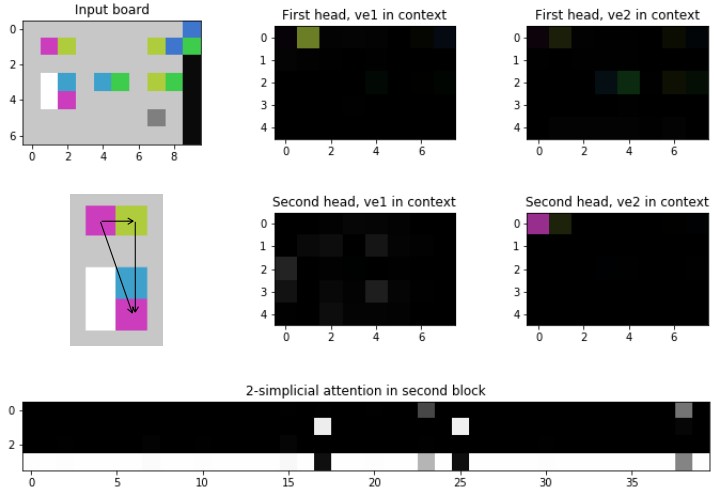

Figure 7: Visualization of 2-simplicial attention in step 18 of an episode.

## 7 DISCUSSION

On general grounds one might expect that in the limit of infinite experience, any reinforcement learning agent with a sufficiently deep neural network will be able to solve any environment, in-

cluding those like bridge BoxWorld that involve higher-order relations between entities. In practice, however, we do not care about the infinite computation limit. In the regime of bounded computation it is reasonable to introduce biases towards learning representations of structures that are found in a wide range of environments that we consider important.

We argue that higher-order relations between entities are an important example of such structures, and that the 2-simplicial Transformer is a natural inductive bias for 3-way interactions between entities. We have given preliminary evidence for the utility of this bias by showing that in the bridge BoxWorld environment the simplicial agent has better performance than a purely relational agent, and that this performance involves in a meaningful way the prediction of 3-way interactions (or 2-simplices). We believe that simplicial Transformers may be useful for any problem in which higher-order relations between entities are important.

The long history of interactions between logic and algebra is a natural source of inspiration for the design of inductive biases in deep learning. In this paper we have exhibited one example: Boole's idea, that relationships between entities can be modeled by multiplication in an algebra, may be realised in the context of deep learning as an augmentation to the Transformer architecture using Clifford algebras of spaces of representations.

ACKNOWLEDGMENTS

We acknowledge support from the Nectar cloud at the University of Melbourne and GCP research credits. DM thanks Paul Middlebrooks for his excellent podcast "Brain Inspired", where he first learned about cognitive maps.

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

APPENDIX

## A  COMPARISON TO THE NTM

The Transformer model and descendents such as the Universal Transformer (Dehghani et al., 2019) can be viewed as general units for computing with learned representations; in this sense they have a similar conceptual role to the Neural Turing Machine (NTM) (Graves et al., 2014) and Differentiable Neural Computer (Graves et al., 2016). As pointed out in (Dehghani et al., 2019, §4) one can view the Transformer as a block of parallel RNNs (one for each entity) which update their hidden states at each time step by attending to the sequence of hidden states of the other RNNs at the previous step. We expand on those remarks here in order to explain the connection between the 2-simplicial Transformer and earlier work in the NLP literature, which is written in terms of RNNs.

We consider a NTM with content-based addressing only and no sharpening. The core of the NTM is an RNN controller with update rule

$$h' = \text{ReLU}(M + Wh + Ux + b) \tag{15}$$

where $W, U, b$ are weight matrices, $x$ is the current input symbol, $h$ is the previous hidden state, $h'$ is the next hidden state and $M$ is the output of the *memory read head*

$$M = \sum_{j=1}^{N} \text{softmax}(K[q, M_1], \ldots, K[q, M_N])_j M_j \tag{16}$$

where there are $N$ memory slots containing $M_1, \ldots M_N$, $q$ is a query generated from the hidden state of the RNN by a weight matrix $q = Zh$, and $K[u, v] = (u \cdot v)/(\|u\|\|v\|)$. We omit the mechanism for writing to the memory here, since it is less obvious how that relates to the Transformer; see (Graves et al., 2014, §3.2). Note that while we can view $M_j$ as the "hidden state" of memory slot $j$, the controller's hidden state and the hidden states of the memory slots play asymmetric roles, since the former is updated with a feedforward network at each time step, while the latter is not.

The Transformer with shared transition functions between layers is analogous to a NTM with this asymmetry removed: there is no longer a separate recurrent controller, and every memory slot is updated with a feedforward network in each timestep. To explain, view the entity representations $e_1, \ldots, e_N$ of the Transformer as the hidden states of $N$ parallel RNNs. The new representation is

$$e_i' = \text{LayerNorm}(g_\theta(A) + e_i) \tag{17}$$

where the attention term is

$$A = \sum_{j=1}^{N} \text{softmax}(q_i \cdot k_1, \ldots, q_i \cdot k_N) v_j \tag{18}$$

and $q_i = Ze_i$ is a query vector obtained by a weight matrix from the hidden state, the $k_j = Ke_j$ are key vectors and $v_j = Ve_j$ is the value vector. Note that in the Transformer the double role of $M_j$

in the NTM has been replaced by two separate vectors, the key and value, and the cosine similarity $K[-, -]$ has been replaced by the dot product.

Having now made the connection between the Transformer and RNNs, we note that the second-order RNN (Giles et al., 1989; Pollack, 1991; Goudreau et al., 1994; Giles et al., 1991) and the similar multiplicative RNN (Sutskever et al., 2011; Irsoy & Cardie, 2015) have in common that the update rule for the hidden state of the RNN involves a term $V(x \otimes h)$ which is a linear function of the tensor product of the current input symbol $x$ and the current hidden state $h$. One way to think of this is that the weight matrix $V$ maps inputs $x$ to linear operators on the hidden state. In (Socher et al., 2013) the update rule contains a term $V(e_1 \otimes e_2)$ where $e_1, e_2$ are entity vectors, and this is directly analogous to our construction.

## B    CONNECTION TO HOPFIELD NETWORKS

The continuous Hopfield network (Hopfield, 1982) (MacKay, 2003, Ch.42) with $N$ nodes updates in each timestep a sequence of vectors $\{e_i\}_{i=1}^N$ by the rules

$$e_i' = \tanh\left[\eta \sum_j (e_i \cdot e_j) e_j\right] \tag{19}$$

for some parameter $\eta$. The Transformer block may therefore be viewed as a refinement of the Hopfield network, in which the three occurrences of entity vectors in (19) are replaced by query, key and value vectors $W^Q e_i, W^K e_j, W^V e_j$ respectively, the nonlinearity is replaced by a feedforward network with multiple layers, and the dynamics are stabilised by layer normalisation. The initial representations $e_i$ also incorporate information about the underlying lattice, via the positional embeddings.

The idea that the structure of a sentence acts to *transform* the meaning of its parts is due to Frege (Frege, 1892) and underlies the denotational semantics of logic. From this point of view the Transformer architecture is an inheritor both of the logical tradition of denotational semantics, and of the statistical mechanics tradition via Hopfield networks.

## C    CLIFFORD ALGEBRA

The volume of an $n$-simplex in $\mathbb{R}^n$ with vertices at $0, v_1, \ldots, v_n$ is

$$\mathrm{Vol}_n = \left| \frac{1}{n!} \det(v_1, \ldots, v_n) \right|$$

which is $\frac{1}{n!}$ times the volume of the $n$-dimensional parallelotope which shares $n$ edges with the $n$-simplex. In our applications the space of representations $H$ is high dimensional, but we wish to speak of the volume of $k$-simplices for $k < \dim(H)$ and use those volumes to define the coefficients of our simplicial attention. The theory of Clifford algebras (Hestenes, 2002) is one appropriate framework for such calculations.

Let $H$ be an inner product space with pairing $(v, w) \mapsto v \cdot w$. The Clifford algebra $\mathrm{Cl}(H)$ is the associative unital $\mathbb{R}$-algebra generated by the vectors $v \in H$ with relations

$$vw + wv = 2(v \cdot w) \cdot 1.$$

The canonical $k$-linear map $H \longrightarrow \mathrm{Cl}(H)$ is injective, and since $v^2 = \|v\|^2 \cdot 1$ in $\mathrm{Cl}(H)$, any nonzero vector $v \in H$ is a unit in the Clifford algebra. While as an algebra $\mathrm{Cl}(H)$ is only $\mathbb{Z}_2$-graded, there is nonetheless a $\mathbb{Z}$-grading of the underlying vector space which can be defined as follows: let $\{e_i\}_{i=1}^n$ be an orthonormal basis of $H$, then the set

$$\mathcal{B} = \left\{ e_{i_1} \cdots e_{i_m} \right\}_{i_1 < \cdots < i_m}$$

is a basis for $\mathrm{Cl}(H)$, with $m$ ranging over the set $\{0, \ldots, n\}$. If we assign the basis element $e_{i_1} \cdots e_{i_m}$ the degree $m$, then this determines a $\mathbb{Z}$-grading $[-]_k$ of the Clifford algebra which is easily checked to be independent of the choice of basis.

**Definition C.1.** $[A]_k$ denotes the homogeneous component of $A \in \mathrm{Cl}(H)$ of degree $k$.

**Example C.2.** Given $a, b, c \in H$ we have $[ab]_0 = a \cdot b$, $[ab]_2 = a \wedge b$ and

$$[abc]_1 = (a \cdot b)c - (a \cdot c)b + (b \cdot c)a, \qquad [abc]_3 = a \wedge b \wedge c. \tag{20}$$

There is an operation on elements of the Clifford algebra called *reversion* in geometric algebra (Hestenes, 2002, p.45) which arises as follows: the opposite algebra $\mathrm{Cl}(H)^{\mathrm{op}}$ admits a $k$-linear map $j : H \longrightarrow \mathrm{Cl}(H)^{\mathrm{op}}$ with $j(v) = v$ which satisfies $j(v)j(w) + j(w)j(v) = 2(v \cdot w) \cdot 1$, and so by the universal property there is a unique morphism of algebras

$$(-)^\dagger : \mathrm{Cl}(H) \longrightarrow \mathrm{Cl}(H)^{\mathrm{op}}$$

which restricts to the identity on $H$. Note $(v_1 \cdots v_k)^\dagger = v_k^\dagger \cdots v_1^\dagger$ for $v_1, \ldots, v_k \in H$ and $(-)^\dagger$ is homogeneous of degree zero with respect to the $\mathbb{Z}$-grading. Using this operation we can define the magnitude (Hestenes, 2002, p.46) of any element of the Clifford algebra.

**Definition C.3.** The *magnitude* of $A \in \mathrm{Cl}(H)$ is $|A| = \sqrt{[A^\dagger A]_0}$.

For vectors $v_1, \ldots, v_k \in H$,

$$|v_1 \cdots v_k|^2 = [v_k \cdots v_1 v_1 \cdots v_k]_0 = \|v_1\|^2 \cdots \|v_k\|^2 \tag{21}$$

and in particular for $v \in H$ we have $|v| = \|v\|$.

**Lemma C.4.** *Set $n = \dim(H)$. Then for $A \in \mathrm{Cl}(H)$ we have*

$$|A|^2 = \sum_{i=0}^{n} |[A]_i|^2.$$

*Proof.* See (Hestenes, 2002, Chapter 2 (1.33)). $\square$

**Example C.5.** For $a, b, c \in H$ the lemma gives

$$\|a\|^2 \|b\|^2 \|c\|^2 = |[abc]_1|^2 + |[abc]_3|^2 = \|(a \cdot b)c - (a \cdot c)b + (b \cdot c)a\|^2 + |a \wedge b \wedge c|^2$$

and hence

$$\|(a \cdot b)c - (a \cdot c)b + (b \cdot c)a\|^2 = \|a\|^2 \|b\|^2 \|c\|^2 - |a \wedge b \wedge c|^2.$$

**Remark C.6.** Given vectors $v_1, \ldots, v_k \in H$ the wedge product $v_1 \wedge \cdots \wedge v_k$ is an element in the exterior algebra $\bigwedge H$. Using the chosen basis $\mathcal{B}$ we can identify the underlying vector space of $\mathrm{Cl}(H)$ with $\bigwedge H$ and using this identification (set $v_i = \sum_j \lambda_{ij} e_j$)

$$
\begin{aligned}
[v_1 \cdots v_k]_k &= \left[ \left( \sum_{j_1=1}^{n} \lambda_{1 j_1} e_{j_1} \right) \cdots \left( \sum_{j_k=1}^{n} \lambda_{k j_k} e_{j_k} \right) \right]_k \\
&= \sum_{j_1, \ldots, j_k} \lambda_{1 j_1} \cdots \lambda_{k j_k} [e_{j_1} \cdots e_{j_k}]_k \\
&= \sum_{\substack{1 \le j_1 < \cdots < j_k \le n \\ \sigma \in S_k}} \lambda_{1 j_{\sigma(1)}} \cdots \lambda_{k j_{\sigma(k)}} [e_{j_{\sigma(1)}} \cdots e_{j_{\sigma(k)}}]_k \\
&= \sum_{\substack{1 \le j_1 < \cdots < j_k \le n \\ \sigma \in S_k}} (-1)^{|\sigma|} \lambda_{1 j_{\sigma(1)}} \cdots \lambda_{k j_{\sigma(k)}} e_{j_1} \cdots e_{j_k} \\
&= v_1 \wedge \cdots \wedge v_k
\end{aligned}
$$

where $S_k$ is the permutation group on $k$ letters. That is, the top degree piece of $v_1 \cdots v_k$ in $\mathrm{Cl}(H)$ is always the wedge product. It is then easy to check that the squared magnitude of this wedge product is

$$|[v_1 \cdots v_k]_k|^2 = \sum_{1 \le j_1 < \cdots < j_k \le n} \left( \sum_{\sigma \in S_k} \lambda_{1 j_{\sigma(1)}} \cdots \lambda_{k j_{\sigma(k)}} \right)^2. \tag{22}$$

The term in the innermost bracket is the determinant of the $k \times k$ submatrix with columns $\mathbf{j} = (j_1, \ldots, j_k)$ and in the special case where $k = n = \dim(H)$ we see that the squared magnitude is just the square of the determinant of the matrix $(\lambda_{ij})_{1 \le i, j \le n}$.

The wedge product of $k$-vectors in $H$ can be thought of as an oriented $k$-simplex, and the magnitude of this wedge product in the Clifford algebra computes the volume.

**Definition C.7.** The *volume* of a $k$-simplex in $H$ with vertices $0, v_1, \ldots, v_k$ is

$$\mathrm{Vol}_k = \frac{1}{k!} \left| [v_1 \cdots v_k]_k \right|. \tag{23}$$

**Definition C.8.** Given $v_1, \ldots, v_k \in H$ the *$k$-fold unsigned scalar product* is

$$\langle v_1, \ldots, v_k \rangle = \sqrt{\sum_{i=0}^{k-1} \left| [v_1 \cdots v_k]_i \right|^2}. \tag{24}$$

By Lemma C.4 and (21) we have

$$\langle v_1, \ldots, v_k \rangle^2 = \|v_1\|^2 \cdots \|v_k\|^2 - (k!)^2 \, \mathrm{Vol}_k^2 \tag{25}$$

which gives the desired generalisation of the equations in Figure 2.

**Example C.9.** For $k = 2$ the unsigned scalar product is the absolute value of the dot product, $\langle a, b \rangle = |a \cdot b|$. For $k = 3$ we obtain the formulas of Definition 2.2, from which it is easy to check that

$$\langle a, b, c \rangle = \|a\| \|b\| \|c\| \sqrt{\cos^2 \theta_{ab} + \cos^2 \theta_{bc} + \cos^2 \theta_{ac} - 2 \cos \theta_{ab} \cos \theta_{ac} \cos \theta_{bc}} \tag{26}$$

where $\theta_{ac}, \theta_{ab}, \theta_{ac}$ are the angles between $a, b, c$. The geometry of the three-dimensional case is more familiar: if $\dim(H) = 3$ then $|[abc]_3|$ is the absolute value of the determinant by (22), so that $\mathrm{Vol}_3 = \frac{1}{6} |a \cdot (b \times c)|$ is the usual formula for the volume of the 3-simplex. Recall that $|a \cdot (b \times c)| = \|a\| \|b\| \|c\| |\sin(\theta_{bc})| |\cos(\phi)|$ where $\phi$ is the angle between $a$ and the cross product $b \times c$. Hence, in this case the scalar triple product is

$$\langle a, b, c \rangle = \|a\| \|b\| \|c\| \sqrt{1 - \sin^2(\theta_{ab}) \cos^2(\phi)}. \tag{27}$$

With these formulas in mind the geometric content of the following lemma is clear:

**Lemma C.10.** *Let $v_1, \ldots, v_k \in H$. Then*

*(i)* $0 \leq \langle v_1, \ldots, v_k \rangle \leq \|v_1\| \cdots \|v_k\|$.

*(ii) If the $v_i$ are all pairwise orthogonal then $\langle v_1, \ldots, v_k \rangle = 0$.*

*(iii) The set $\{v_1, \ldots, v_k\}$ is linearly dependent if and only if $\langle v_1, \ldots, v_k \rangle = \|v_1\| \cdots \|v_k\|$.*

*(iv) For any $\sigma \in S_k$ we have $\langle v_1, \ldots, v_k \rangle = \langle v_{\sigma(1)}, \cdots, v_{\sigma(k)} \rangle$.*

*(v) For $\lambda_1, \ldots, \lambda_k \in \mathbb{R}$, we have*

$$\langle \lambda_1 v_1, \ldots, \lambda_k v_k \rangle = |\lambda_1| \cdots |\lambda_k| \langle v_1, \ldots, v_k \rangle.$$

*Proof.* (i) is obvious from (24), (25). For (ii) note that

$$v_1 \wedge \cdots \wedge v_k = \frac{1}{k!} \sum_{\sigma \in S_k} (-1)^{|\sigma|} v_{\sigma(1)} \cdots v_{\sigma(k)} \tag{28}$$

and hence if the $v_i$ are pairwise orthogonal, and therefore anticommute in $\mathrm{Cl}(H)$, we have $v_1 \wedge \cdots \wedge v_k = v_1 \cdots v_k$. But the left hand side is homogeneous of degree $k$, so this means that $[v_1 \cdots v_k]_i = 0$ for $i < k$ and hence that $\langle v_1, \ldots, v_k \rangle = 0$. The property (iii) is a standard property of wedge products. Finally, (iv) is clear from (25) and (v) is clear since $|\lambda A| = |\lambda| |A|$ for any $A \in \mathrm{Cl}(H)$. $\square$

For more on simplicial methods in the context of geometric algebra see (Sobczyk, 1992; Macdonald, 2017).

## D    TIME ADJUSTED PERFORMANCE

Experiments were conducted either on the Google Cloud Platform with a single head node with 12 virtual CPUs and one NVIDIA Tesla P100 GPU and 192 additional virtual CPUs spread over two pre-emptible worker nodes, or on the University of Melbourne Nectar research cloud with a single head node with 12 virtual CPUs and two NVIDIA Tesla K80 GPUs, and 222 worker virtual CPUs.

The experiments in the original BoxWorld paper (Zambaldi et al., 2019) contain an unreported cap on timesteps per episode (an *episode horizon*) of 120 timesteps (Raposo, 2019). We have chosen to run our experiments without an episode horizon, and since this means our reported sample complexities diverge substantially from the original paper (some part of which it seems reasonable to attribute to the lack of horizon) it is necessary to justify this choice.

When designing an architecture for deep reinforcement learning the goal is to reduce the expected generalisation error (Goodfellow et al., 2016, §8.1.1) with respect to some class of similar environments. Although this class is typically difficult to specify and is often left implicit, in our case the class includes a range of visual logic puzzles involving spatial navigation, which can be solved without memory[2]. A learning curriculum undermines this goal, by making our expectations of generalisation conditional on the provision of a suitable curriculum, whose existence for a given member of the problem class may not be clear in advance. The episode horizon serves as a *de facto* curriculum, since early in training it biases the distribution of experience rollouts towards the initial problems that an agent has to solve (e.g. learning to pick up the loose key). In order to avoid compromising our ability to expect generalisation to similar puzzles which do not admit such a useful curriculum, we have chosen not to employ an episode horizon. Fortunately, the relational agent performs well even without a curriculum on the original BoxWorld, as our results show.

In Figure 6 of Section 5, the horizontal axis was environment steps. However, since the simplicial agent has a more complex model, each environment step takes longer to execute and the gradient descent steps are slower. In a typical experiment run on the GCP configuration, the training throughput of the relational agent is $1.9 \times 10^4$ environment frames per second (FPS) and that of the simplicial agent is $1.4 \times 10^4$ FPS. The relative performance gap decreases as the GPU memory and the number of IMPALA workers are increased, and this is consistent with the fact that the primary performance difference appears to be the time taken to compute the gradients (35ms vs 80ms). In Figure 8 we give the time-adjusted performance of the simplicial agent (the graph for the relational agent is as before) where the $x$-axis of the graph of the simplicial agent is scaled by $1.9/1.4$.

In principle there is no reason for a significant performance mismatch: the 2-simplicial attention can be run in parallel to the ordinary attention (perhaps with two iterations of the 1-simplicial attention per iteration of the 2-simplicial attention) so that with better engineering it should be possible to reduce this gap.

## E    HYPERPARAMETERS

| Hyperparameter | Value |
|---|---|
| IMPALA entropy | $5 \times 10^{-3}$ |
| Discount factor $\gamma$ | 0.99 |
| Unroll length | 40 timesteps |
| Batch size | 1280 timesteps |
| Learning rate | $2 \times 10^{-4}$ |
| RMSProp momentum | 0 |
| RMSProp $\varepsilon$ | 0.1 |
| RMSProp decay | 0.99 |

Table 1: Hyperparameters for agent training.

---

[2]The bridge is the unique box both of whose colours appear three times on the board. However, this is not a reliable strategy for detecting bridges for an agent without memory, because once the agent has collected some of the keys on the board, some of the colours necessary to make this deduction may no longer be present.

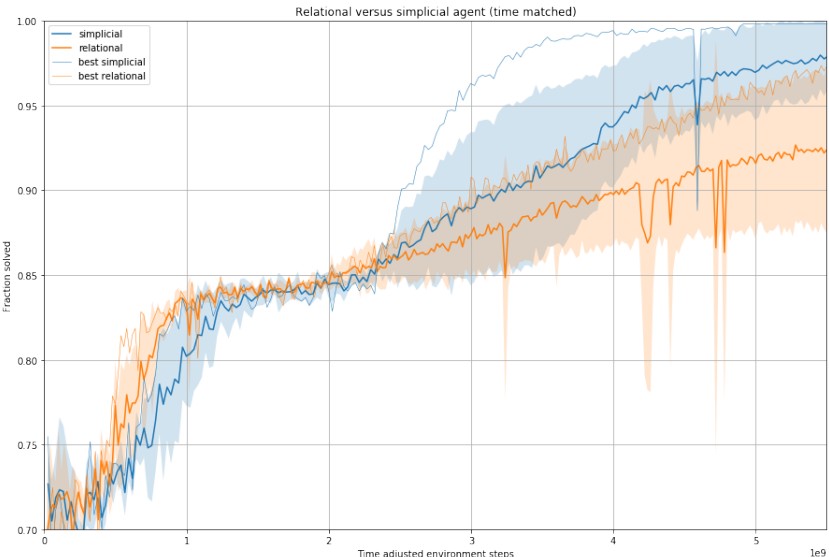

Figure 8: Training curve of mean relational and simplicial agents on bridge BoxWorld, with time-adjusted $x$-axis for the simplicial agent.

## F  FURTHER ANALYSIS

Our experiments involve only a small number of virtual entities, and a small number of iterations of the Transformer block: it is possible that for large numbers of virtual entities and iterations, our choices of layer normalisation are not optimal. Our aim was to test the viability of the simplicial Transformer starting with the minimal configuration, so we have also not tested multiple heads of 2-simplicial attention. Deep reinforcement learning is notorious for poor reproducibility (Henderson et al., 2017), and in an attempt to follow the emerging best practices we are releasing our agent and environment code, trained agent weights, and training notebooks (Clift et al., 2019).

The training runs for the relational and simplicial agents are shown in Figure 9 and Figure 10 respectively.

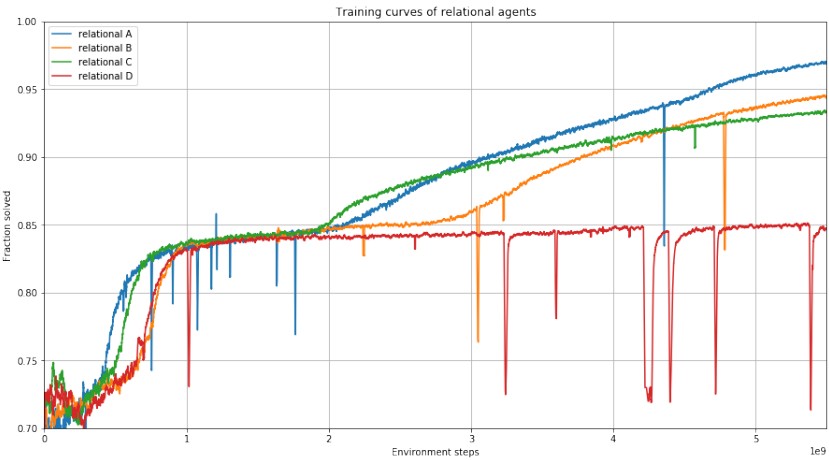

Figure 9: Training curves for the relational agent on bridge BoxWorld.

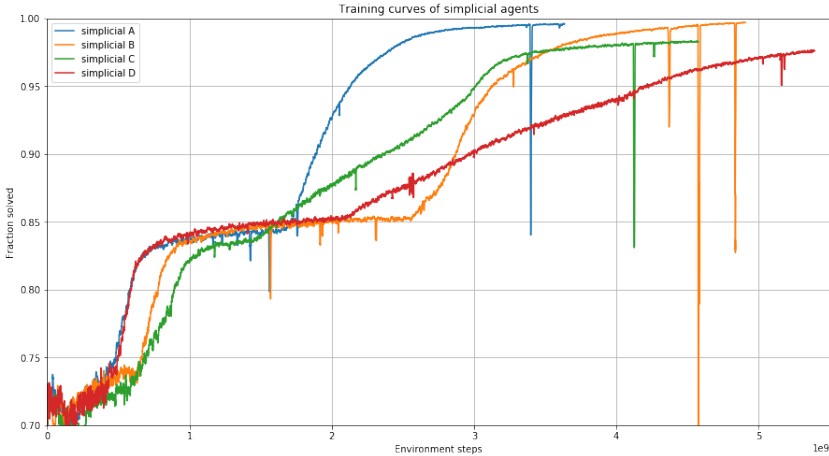

Figure 10: Training curves for the simplicial agent on bridge BoxWorld.

## F.1 ATTENTION

In this Appendix we provide further details relating to the analysis of the attention of the trained simplicial agent in Section 6. Across our four trained simplicial agents, the roles of the virtual entities and heads vary: the following comments are all in the context of the best simplicial agent (simplicial agent A of Figure 10) but we observe similar patterns in the other trials.

### F.1.1 1-SIMPLICIAL ATTENTION OF STANDARD ENTITIES

The standard entities are now indexed by $0 \leq i \leq 39$ and virtual entities by $i = 40, 41$. In the first iteration of the 2-simplicial Transformer block, the first 1-simplicial head appears to propagate information about the inventory. At the beginning of an episode the attention of each standard entity is distributed between entities $7, 15, 23, 31$ (the entities in the rightmost column), it concentrates sharply on $7$ (the entity closest to the first inventory slot) after the acquisition of the first loose key, and sharply on $7, 15$ after the acquisition of the second loose key. The second 1-simplicial head seems to acquire the meaning described in (Zambaldi et al., 2019), where tiles of the same colour attend to one another. A typical example is shown in Figure 11. The video of this episode is available online (Clift et al., 2019).

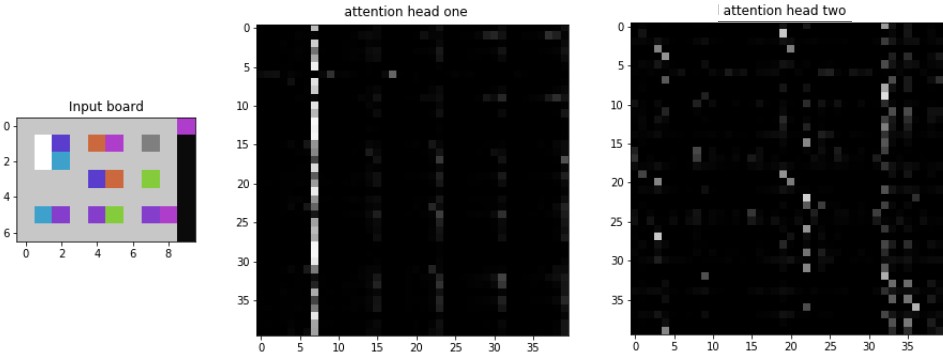

Figure 11: Visualisation of 1-simplicial attention in first Transformer block, between standard entities in heads one and two. The vertical axes on the second and third images are the query index $0 \leq i \leq 39$, the horizontal axes are the key index $0 \leq j \leq 39$.

### F.1.2 2-SIMPLICIAL ATTENTION

The standard entities are updated using 2-simplices in the first iteration of the 2-simplicial Transformer block, but this is not interesting as initially the virtual entities are learned embedding vectors, containing no information about the current episode. So we restrict our analysis to the 2-simplicial attention in the *second* iteration of the Transformer block.

For the analysis, it will be convenient to organise episodes of bridge BoxWorld by their *puzzle type*, which is the tuple $(a, b, c)$ where $1 \leq a \leq 3$ is the solution length, $1 \leq b \leq a$ is the *bridge source* and $a + 1 \leq c \leq 2a$ is the *bridge target*, with indices increasing with the distance from the gem. The episodes in Figures 4 and 7 have type $(3, 2, 5)$.

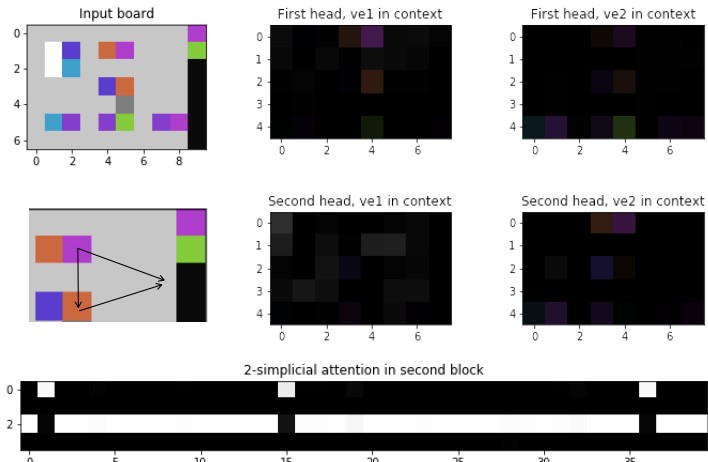

Figure 12: Visualisation of the 2-simplicial attention in the second Transformer block in step 13 of an episode of puzzle type $(3, 3, 5)$. Entity 1 is the top lock on the Gem, 15 is associated with the inventory, 36 is the lock directly below the player. Shown is a 2-simplex with target 15.

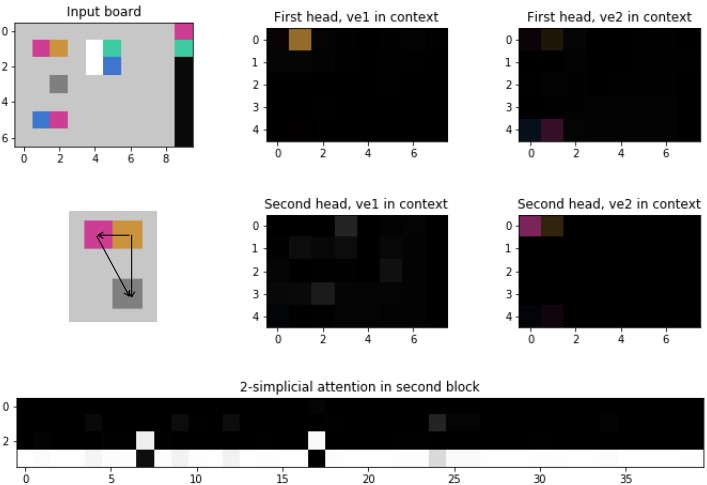

Figure 13: Visualisation of the 2-simplicial attention in the second Transformer block in step 29 of an episode of puzzle type $(3, 3, 5)$. Entity 7 is associated with the inventory, 17 is the player. Shown is a 2-simplex with target 17.

To give more details we must first examine the content of the virtual entities after the first iteration, which is a function of the 1-simplicial attention of the virtual entities in the first iteration. In Figures 7, 12, 13 we show these attention distributions multiplied by the pixels in the region $[1, R - 2] \times$

$[1, C - 1]$ of the original board, in the second and third columns of the second and third rows.[3] Let $f_1 = e_{40}$ and $f_2 = e_{41}$ denote the initial representations of the first and second virtual entities, before the first iteration. We use the index $z \in \{1, 2\}$ to stand for a virtual entity. In the first iteration the representations are updated by (14) to

$$f'_z = \text{LayerNorm}\left(g_\theta\left[\left\{\sum_\alpha a^z_\alpha v_\alpha\right\} \oplus \left\{\sum_\alpha b^z_\alpha v_\alpha\right\}\right] + f_z\right) \tag{29}$$

where the sum is over all entities $\alpha$, the $a^z_\alpha$ are the attention coefficients of the first 1-simplicial head and the coefficients $b^z_\alpha$ are the attention of the second 1-simplicial head. Writing $\mathbf{0}_1, \mathbf{0}_2$ for the zero vector in $H^1_1, H^1_2$ respectively, this can be written as

$$f'_z = \text{LayerNorm}\left(g_\theta\left[\sum_\alpha a^z_\alpha(v_\alpha \oplus \mathbf{0}_2) + \sum_\alpha b^z_\alpha(\mathbf{0}_1 \oplus v_\alpha)\right] + f_z\right). \tag{30}$$

For a query entity $i$ the vector propagated by the 2-simplicial part of the second iteration has the following terms, where $\widetilde{B} = B \circ (W^U \otimes W^U)$

$$A^i_{1,1}\widetilde{B}(f'_1 \otimes f'_1) + A^i_{1,2}\widetilde{B}(f'_1 \otimes f'_2) + A^i_{2,1}\widetilde{B}(f'_2 \otimes f'_1) + A^i_{2,2}\widetilde{B}(f'_2 \otimes f'_2). \tag{31}$$

Here $A^i_{j,k}$ is the 2-simplicial attention with logits $\langle p_i, l^1_j, l^2_k \rangle$ associated to $(i, j, k)$.

The tuple $(A^i_{1,1}, A^i_{1,2}, A^i_{2,1}, A^i_{2,2})$ is the $i$th column in our visualisations of the 2-simplicial attention, so in the situation of Figure 7 with $i = 25$ we have $A^{25}_{1,2} \approx 1$ and hence the output of the 2-simplicial head used to update the entity representation of the bottom lock on the Gem is approximately $\widetilde{B}(f'_1 \otimes f'_2)$. If we ignore the layer normalisation, feedforward network and skip connection in (30) then $f'_1 \approx v_1 \oplus \mathbf{0}_2$ and $f'_2 \approx \mathbf{0}_1 \oplus v_0$ so that the output of the 2-simplicial head with target $i = 25$ is approximately

$$\widetilde{B}((v_1 \oplus \mathbf{0}_2) \otimes (\mathbf{0}_1 \oplus v_0)). \tag{32}$$

Following Boole (Boole, 1847) and Girard (Girard, 1987) it is natural to read the "product" (32) as a conjunction (consider together the entity 1 and the entity 0) and the sum in (31) as a disjunction. An additional layer normalisation is applied to this vector, and the result is concatenated with the incoming information for entity 25 from the 1-simplicial attention, before all of this is passed through (12) to form $e'_{25}$.

Given that the output of the 2-simplicial head is the only nontrivial difference between the simplicial and relational agent (with a transformer depth of two, the first 2-simplicial Transformer block only updates the standard entities with information from embedding vectors) the performance differences reported in Figure 6 suggest that this output is informative about avoiding bridges.

### F.2    THE PLATEAU

In the training curves of the agents of Figure 9 and Figure 10 we observe a common plateau at a win rate of $0.85$. In Figure 14 we show the per-puzzle win rate of simplicial agent A and relational agent A, on $(1, 1, 2)$ puzzles. These graphs make clear that the transition of both agents to the plateau at $0.85$ is explained by solving the $(1, 1, 2)$ type (and to a lesser degree by progress on all puzzle types with $b = 1$). In Figure 14 and Figure 15 we give the per-puzzle win rates for a small sample of other puzzle types. Shown are the mean and standard deviation of 100 runs across various checkpoints of simplicial agent A and relational agent A.

## G    LARGE EPSILON RMSPROP

As originally presented in (Tieleman & Hinton, 2012) the optimisation algorithm RMSProp is a mini-batch version of Rprop, where instead of dividing by a different number in every mini-batch

---

[3]For visibility in print the 1-simplicial attention of the virtual entities in these figures has been sharpened, by multiplying the logits by 2. The 2-simplicial attention and 1-simplicial attention of standard entities have not been sharpened. In this connection, we remark that in Figure 7 there is one entity whose unsharpened attention coefficient for the first virtual entity in the first head is more than one standard deviation above the mean, and there are two such entities for the second virtual entity and second head.

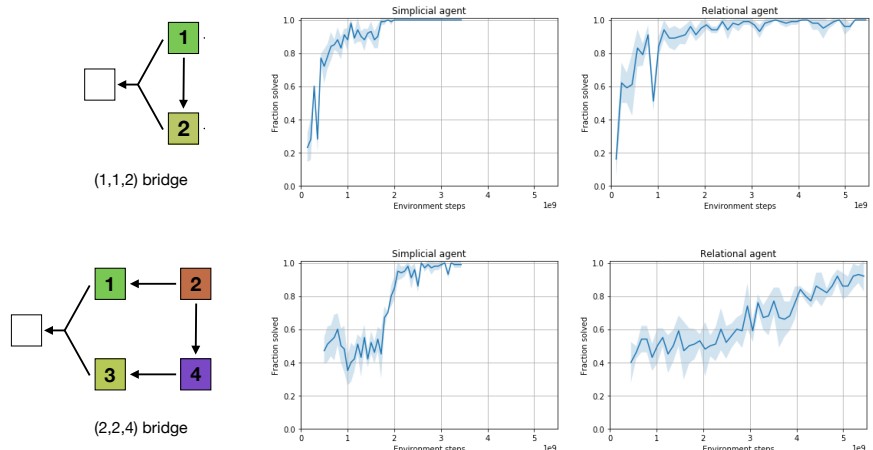

Figure 14: Simplicial and relational agent win rate on puzzle types $(1, 1, 2), (2, 2, 4)$.

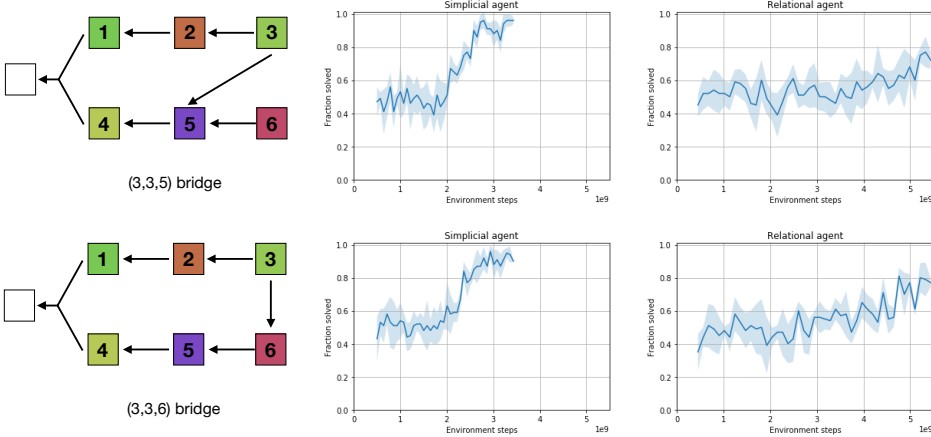

Figure 15: Simplicial and relational agent win rate on puzzle types $(3, 3, 5), (3, 3, 6)$.

(namely, the absolute value of the gradient) we force this number to be similar for adjacent mini-batches by keeping a moving average of the square of the gradient. In more detail, one step Rprop is computed by the algorithm

$$r_i \leftarrow g_i^2$$
$$x_i \leftarrow x_i - \frac{\kappa g_i}{\sqrt{r_i + \varepsilon}}$$

where $\kappa$ is the learning rate, $x_i$ is a weight, $g_i$ is the associated gradient and $\varepsilon$ is a small constant (the TensorFlow default value is $10^{-10}$) added for numerical stability. The idea of Rprop is to update weights using only the *sign* of the gradient: every weight is updated by the same absolute amount $\kappa$ in each step, with only the sign $g_i/\sqrt{r_i} = g_i/|g_i|$ of the update varying with $i$. The algorithm RMSprop was introduced as a refinement of Rprop:

$$r_i \leftarrow p r_i + (1 - p) g_i^2$$
$$x_i \leftarrow x_i - \frac{\kappa g_i}{\sqrt{r_i + \varepsilon}}$$

where $p$ is the *decay rate* (in our experiments the value is 0.99). Clearly Rprop is the $p \to 0$ limit of RMSprop. For further background see (Goodfellow et al., 2016, §8.5.2).

In recent years there has been a trend in the literature towards using RMSprop with large values of the hyperparameter $\varepsilon$. For example in (Zambaldi et al., 2019) RMSProp is used with $\varepsilon = 0.1$, which

is also one of the range of values in (Espeholt et al., 2018, Table D.1) explored by population based training (Jaderberg et al., 2017). This "large $\varepsilon$ RMSProp" seems to have originated in (Szegedy et al., 2016, §8). To understand what large $\varepsilon$ RMSProp is doing, let us rewrite the algorithm as

$$r_i \leftarrow pr_i + (1-p)g_i^2$$

$$x_i \leftarrow x_i - \frac{\kappa g_i}{\sqrt{r_i}} \cdot \frac{1}{\sqrt{1 + \varepsilon/r_i}}$$

$$= x_i - \frac{\kappa g_i}{\sqrt{r_i}} S\left[\frac{\sqrt{r_i}}{\sqrt{\varepsilon}}\right]$$

where $S$ is the sigmoid $S(u) = u/\sqrt{1 + u^2}$ which asymptotes to 1 as $u \rightarrow +\infty$ and is well-approximated by the identity function for small $u$. We see a new multiplicative factor $S(\sqrt{r_i/\varepsilon})$ in the optimisation algorithm. Note that $\sqrt{r_i}$ is a moving average of $|g_i|$. Recall the original purpose of Rprop was to update weights using only the sign of the gradient and the learning rate, namely $\kappa g_i/\sqrt{r_i}$. The new $S$ factor in the above reinserts the size of the gradient, but scaled by the sigmoid to be in the unit interval.

In the limit $\varepsilon \rightarrow 0$ we squash the outputs of the sigmoid up near 1 and the standard conceptual description of RMSProp applies. But as $\varepsilon \rightarrow 1$ the sigmoid $S(\sqrt{r_i})$ has the effect that for large stable gradients we get updates of size $\kappa$ and for small stable gradients we get updates of the same magnitude as the gradient. In conclusion, large $\varepsilon$ RMSprop is a form of RMSprop with *smoothed gradient clipping* (Goodfellow et al., 2016, §10.11.1).

## H  Logic and reinforcement learning

It is no simple matter to define *logical reasoning* nor to recognise when an agent (be it an animal or a deep reinforcement learning agent) is employing such reasoning (Mackintosh, 2019; Barrett et al., 2018). We therefore begin by returning to Aristotle, who viewed logic as the study of general patterns by which one could distinguish valid and invalid forms of philosophical argumentation; this study having as its purpose the production of *strategies* for winning such argumentation games (Aristotle, 1984; Smith, 2019; Spade & Hintikka, 2019). In this view, logic involves

- **two players** with one asserting the truth of a proposition and attempting to defend it, and the latter asserting its falsehood and attempting to refute it, and an

- **observer** attempting to learn the general patterns which are predictive of which of the two players will win such a game given some intermediate state.

Suppose we observe over a series of games[4] that a player is following an explicit strategy which has been distilled from general patterns observed in a large distribution of games, and that by following this strategy they almost always win. A component of that explicit strategy can be thought of as logical reasoning to the degree that it consists of rules that are independent of the particulars of the game (Aristotle, 1984, §11.25). The problem of recognising logical reasoning in behaviour is therefore twofold: the strategy employed by a player is typically *implicit*, and even if we can recognise explicit components of the strategy, in practice there is not always a clear way to decide which rules are domain-specific.

In mathematical logic the idea of argumentation games has been developed into a theory of *mathematical proof as strategy* in the game semantics of linear logic (Hyland, 1997) where one player (the *prover*) asserts a proposition $G$ and the other player (the *refuter*) interrogates this assertion.[5]

---

[4]We cannot infer that a behaviour constitutes logical reasoning if we only observe it over the course of a single game. For example, while it may appear that a human proving a statement in mathematics by correctly applying a set of deduction rules is engaged in logical reasoning, this appearance may be false, for if we were to observe one thousand attempts to prove a sample of similar propositions, and in only one attempt was the human able to correctly apply the deduction rules, we would have to retract our characterisation of the behaviour as logical reasoning. The concept is also empty if we insist that it applies only if in *every* such attempt the deduction rules are correctly applied, because human mathematicians make mistakes.

[5]It is possible for the prover to win such an argument without possessing a proof (for instance if $G$ is the disjunct of propositions $A, B$ and the refuter demands a proof of $A$ in a situation where the prover knows a proof of $A$ but not of $B$) but the only strategy guaranteed to win is to play according to a proof.

Consider a reinforcement learning problem (Sutton & Barto, 2018) in which the deterministic environment encodes $G$ together with a multiset of hypotheses $\Gamma$ which are sufficient to prove $G$. Such a pair is called a *sequent* and is denoted $\Gamma \vdash G$. The goal of the agent (in the role of prover) is to synthesise a proof of $G$ from $\Gamma$ through a series of actions. The environment (in the role of refuter) delivers a positive reward if the agent succeeds, and a negative reward if the agent's actions indicate a commitment to a line of proof which cannot possibly succeed. Consider a deep reinforcement learning agent with a policy network parametrised by a vector of weights $\mathbf{w} \in \mathbb{R}^D$ and a sequence of full-episode rollouts of this policy in the environment, each of which either ends with the agent constructing a proof (prover wins) or failing to construct a proof (refuter wins) with the sequent $\Gamma \vdash G$ being randomly sampled in each episode. Viewing these episodes as instances of an argumentation game, the goal of Aristotle's observer is to learn from this data to predict, given an intermediate state of some particular episode, which actions by the prover will lead to success (proof) or failure (refutation). As the reward is correlated with success and failure in this sense, the goal of the observer may be identified with the training objective of the action-value network underlying the agent's policy, and we may identify the triple *player*, *opponent*, *observer* with the triple *agent*, *environment* and *optimisation process*. If this process succeeds, so that the trained agent wins in almost every episode, then by definition the weights $\mathbf{w}$ are an *implicit strategy* for proving sequents $\Gamma \vdash G$.

This leads to the question: is the deep reinforcement learning agent parametrised by $\mathbf{w}$ performing logical reasoning? We would have no reason to deny that logical reasoning is present if we were to find, in the weights $\mathbf{w}$ and dynamics of the agent's network, an isomorphic image of an explicit strategy that we recognise as logically correct. In general, however, it seems more useful to ask *to what degree* the behaviour is governed by logical reasoning, and thus to what extent we can identify an approximate *homomorphic* image in the weights and dynamics of a logically correct explicit strategy. Ultimately this should be automated using "logic probes" along the lines of recent developments in neural network probes (Alain & Bengio, 2016; Koh & Liang, 2017; Nguyen et al., 2016; Shrikumar et al., 2017; Simonyan et al., 2013).

## I  STRATEGIES AND PROOF TREES

The design of the BoxWorld environment was intended to stress the planning and reasoning components of an agent's policy (Zambaldi et al., 2019, p.2) and for this reason it is the underlying logical structure of the environment that is of central importance. To explain the logical structure of BoxWorld and bridge BoxWorld we introduce the following notation: given a colour $c$, we use $C$ to stand for the proposition that a key of this colour is *obtainable*. Each episode expresses its own set of basic facts, or axioms, about obtainability. For instance, a loose key of colour $c$ gives $C$ as an axiom, and a locked box requiring a key of colour $c$ in order to obtain a key of colour $d$ gives an axiom that at first glance appears to be the implication $C \longrightarrow D$ of classical logic. However, since a key may only be used once, this is actually incorrect; instead the logical structure of this situation is captured by the *linear implication $C \multimap D$* of linear logic (Girard, 1987). With this understood, each episode of the original BoxWorld provides in visual form a set of axioms $\Gamma$ such that a strategy for obtaining the Gem is equivalent to a proof of $\Gamma \vdash \mathbb{G}$ in intuitionistic linear logic, where $\mathbb{G}$ stands for the proposition that the Gem is obtainable. There is a general correspondence in logic between strategies and proofs which we recall in Appendix I.

To describe the logical structure of bridge BoxWorld we need to encode the fact that two keys (say a green key *and* a blue key) are required to obtain the Gem. Once again, it is the *linear conjunction* $\otimes$ of linear logic (also called the tensor product) rather than the conjunction of classical logic that properly captures the semantics. The axioms $\Gamma$ encoded in an episode of bridge BoxWorld contain a single formula of the form $X_1 \otimes X_2 \multimap \mathbb{G}$ where $x_1, x_2$ are the colours of the keys on the Gem, and again a strategy is equivalent to a proof of $\Gamma \vdash \mathbb{G}$. In conclusion, the logical structure of the original BoxWorld consists of a fragment of linear logic containing only the connective $\multimap$, while bridge BoxWorld captures a slightly larger fragment containing $\multimap$ and $\otimes$.

Next we explain the correspondence between agent behaviour in bridge BoxWorld and proofs in linear logic. For an introduction to linear logic tailored to the setting of games see (Martens, 2015, Ch.2). Recall that to each colour $c$ we have associated a proposition $C$ which can be read as "the key of colour $c$ is obtainable". If a box $\beta$ appears in an episode of bridge BoxWorld (this includes loose

keys and the box with the Gem) then we assume given a proof $\pi^\beta$ of a sequent associated to the box by the following rules: the sequent $X_\beta \vdash Y_\beta$ associated to a loose key of colour $c$ is $\vdash C$, the sequent associated to an ordinary box with a lock of colour $c$ and containing a key of colour $c'$ is $C \vdash C'$ and the sequent associated to a multiple lock on the Gem, with key colours $c, c'$ is $C \otimes C' \vdash \mathbb{G}$. In the following we identify the box $\beta$ with its associated sequent, and write for example $\pi^{\vdash C}$ for the chosen proof associated to the loose key of colour $c$. The set of premises (or axioms) in an episode of bridge BoxWorld is the multiset $\Gamma$ of proofs $\pi^\beta$ as $\beta$ ranges over all boxes.

**Definition I.1.** Given a formula $A$ (thought of as representing the contents of the inventory) and a box $\beta$ we define the proof $\pi_A^\beta$ to be

$$
\begin{array}{c}
\pi^\beta \\
\vdots \\
\cfrac{\cfrac{A \vdash A \qquad X_\beta \vdash Y_\beta}{A, X_\beta \vdash A \otimes Y_\beta}\ {\otimes R}}{A \otimes X_\beta \vdash A \otimes Y_\beta}\ {\otimes L}
\end{array}
\tag{33}
$$

One can think of this proof as the algorithm which acts to update the contents of the inventory upon opening the box $\beta$.

**Example I.2.** Consider the episode of Figure 4 and suppose that the agent follows the upper solution path and then the lower, obtaining the keys in the following order: $g$ (green), $o$ (orange), $g'$ (dark green), $m$ (magenta), $p$ (purple) and $b$ (blue). Then the proof tree whose computational content matches this behaviour is given by:

$$
\tag{34}
$$

where unlabelled deduction rules are cuts. Cutting this proof tree against the proof $\pi^{G' \otimes B \vdash \mathbb{G}}$ associated to the final box gives the proof encoding the agent's strategy.

This example makes clear the general rule for associating a proof tree to an agent's strategy, as embodied in its behaviour: take the sequence of boxes $\beta_1, \ldots, \beta_N$ opened by the agent together with the state of the inventory $I_1, \ldots, I_N$ at the time of each opening, and cut the corresponding sequence of proofs $\pi_{I_i}^{\beta_i}$ against one another.

## J  MOTIVATION FROM NEUROSCIENCE

The most successful examples of representations in deep learning, those learned by convolutional neural networks, are structured by the scale and translational symmetries of the underlying space (e.g. a two-dimensional Euclidean space for images). It has been suggested that in humans the ability to make rich inferences based on abstract reasoning is rooted in the same neural mechanisms underlying relational reasoning in space (Constantinescu et al., 2016; Epstein et al., 2017; Behrens et al., 2018; Bellmund et al., 2018) and more specifically that abstract reasoning is facilitated by the learning of *structural representations* which serve to organise other learned representations in the same way that space organises the representations that enable spatial navigation (Whittington et al., 2018; Liu et al., 2019).

As a motivating example we take the recent progress on natural language tasks based on the Transformer architecture (Vaswani et al., 2017) which simultaneously learns to represent both entities (typically words) and relations between entities (for instance the relation between "cat" and "he"

in the sentence "There was a cat and he liked to sleep"). These representations of relations take the form of query and key vectors governing the passing of messages between entities; messages update entity representations over several rounds of computation until the final representations reflect not just the meaning of words but also their context in a sentence. There is some evidence that the geometry of these final representations serve to organise word representations in a syntax tree, which could be seen as the appropriate analogue to two-dimensional space in the context of language (Hewitt & Manning, 2019).

The Transformer may therefore be viewed as an inductive bias for learning structural representations which are *graphs*, with entities as vertices and relations as edges. While a graph is a discrete mathematical object, there is a naturally associated topological space which is obtained by gluing 1-simplices (copies of the unit interval) indexed by edges along 0-simplices (points) indexed by vertices. There is a general mathematical notion of a *simplicial set* which is a discrete structure containing a set of $n$-simplices for all $n \geq 0$ together with an encoding of the incidence relations between these simplices. Associated to each simplicial set is a topological space, obtained by gluing together vertices, edges, triangles (2-simplices), tetrahedrons (3-simplices), and so on, according to the instructions contained in the simplicial set. Following the aforementioned works in neuroscience (Constantinescu et al., 2016; Epstein et al., 2017; Behrens et al., 2018; Bellmund et al., 2018; Whittington et al., 2018; Liu et al., 2019) and their emphasis on spatial structure, it is natural to ask if a *simplicial* inductive bias for learning structural representations can facilitate abstract reasoning. This question partly motivated the developments in this paper.

