# OpenReview forum: "Logic and the 2-Simplicial Transformer"
_ICLR.cc/2020/Conference — Accept (Poster)_

### Official Review · AnonReviewer1 · 2019-10-23
**Official Blind Review #1**

**Rating:** 3

**Review:**

This paper is an extension of the Transformer Algorithm used to solve sequential problems such as in NLP and games (such as the BoxWorld environment from Zambaldi et al.), stating that the Transformer Algorithm is an inductive bias for learning structural representations.
The authors argue that, the use of simplicial inductive biases over the normal relational inductive bias (provided by the Transformer algorithm) may be a better way to achieve abstract learning; in fact they argue that the “relational inductive bias” used in the normal Transformer is just a 1-simplicial Transformer, thus the more complex topological space provided by the 2-simplicial Transformer should generate better results than the 1-simplicial Transformer.
While technically sound  it’s not obvious whether  this implementation is indeed better than the normal.

First of all the significance of the results are heavily determined by the fact that the authors just displayed the results of the proposed algorithm against the “normal Transformer”, using a modified version of the BoxWorld. Moreover  in the paper it is stated that, they tested the new algorithm with the original BoxWorld environment, but they omitted it as the proposed algorithm excelled over the old one in such environments. Also the relational induction algorithm used in this paper does not seem to be the same as the one used by Zambaldi et al.

Second, Im concerned about the explanation made where they try to justify that the use of the bridge-BoxWorld which is argued to provide a more complex logical structure because of the use of linear logic-formulas using more connectives. In other words, the question arises as to whether the authors thought that a more complex logical structure would i mply a more complex environment, thus justifying the use of the bridge-BoxWorld.

Finally, it is worth reiterating that, although the use of simplices and simplicial complexes is a very interesting idea, especially to provide a mathematical explanation of how attention works in the Transformer, it’s not entirely obvious  whether the implementation of the 2-simplicity Transformer really presents an improvement over the original Transformer, in the learning of abstract representation.

**Experience Assessment:**

I do not know much about this area.

**Review Assessment: Checking Correctness Of Derivations And Theory:**

I did not assess the derivations or theory.

**Review Assessment: Checking Correctness Of Experiments:**

I did not assess the experiments.

**Review Assessment: Thoroughness In Paper Reading:**

I made a quick assessment of this paper.

---

> ### Author Response · Authors · 2019-11-12
> **Response to Review #1**
>
> Thank you for taking the time to review our paper. You agree that a simplicial inductive bias is an interesting idea, but are concerned that this idea does not lead to a genuine improvement over the original Transformer. We agree that an absolutely conclusive proof of this would involve testing our simplicial inductive bias in many RL environments, as well as other domains such as NLP, and we hope to pursue this in future work. However, our paper nonetheless contains clear evidence that the simplicial Transformer is an improvement over the standard Transformer, as an inductive bias in an RL agent, along at least two axes: pure performance, and richness of representations.
>
> Regarding performance: when developing a new RL algorithm that is proposed to increase performance, the first step is to test this new algorithm against the existing algorithm in a toy environment taken from a similar class of environments used to test the existing algorithm. This was done and as you can see in Figures 5, 7, 8, 9, 13 and 14, our architecture significantly outperforms the standard Transformer for this particular reasoning task. We agree with Reviewer #3 that "it is right to start testing a new architecture by showing it can indeed do what it is designed to do; further tests showing that what it is designed to do is of general utility are a second step".
>
> Regarding representations: our results show that the agent has learned to use the 2-simplicial component, and that these simplicial representations are more complex and interesting than the relational ones. Please see the newly added Section 6 which is a summary of our analysis from the original appendix.
>
> Finally, while we stand behind our experimental results, we view our core contribution as theoretical: we derive a natural form of higher-dimensional attention by introducing Clifford algebras into deep learning, and show how to incorporate this into a sufficiently stable architecture to form the core of a RL agent. We hope that the updated revision better reflects this.
>
> >>Moreover in the paper it is stated that, they tested the new algorithm with the original BoxWorld environment, but they omitted it as the proposed algorithm excelled over the old one in such environments. Also the relational induction algorithm used in this paper does not seem to be the same as the one used by Zambaldi et al.
>
> We chose the best performant version of the relational algorithm for the fairest possible comparison to the simplicial agent. This involved a different arrangement of layer normalizations only. If we had used the precise relational architecture specified in Zambaldi et al, the gap between the simplicial and relational agents would have been wider.
>
> >>In other words, the question arises as to whether the authors thought that a more complex logical structure would imply a more complex environment, thus justifying the use of the bridge-BoxWorld.
>
> The connection to linear logic was motivational for us, and perhaps interesting for a reader with a logical background, but it is not necessary to believe in this connection in order to believe that bridge BoxWorld is a more complex environment than the original BoxWorld. Empirically bridge BoxWorld puzzles are harder to solve, both for us (the authors) and our agents, and the natural hypothesis as to why these puzzles are harder to solve is that they involve conjunctive reasoning which is absent from the original BoxWorld. This hypothesis is supported both by introspection (try to solve the puzzles and see what general strategy you arrive at) and by inspection of the learned strategy of the agent via their attention distributions.
>
> Having said that, we accept that the emphasis given to logic in the original introduction is for many readers out of proportion to its actual importance in the body of the paper. To rectify this we have moved this material into Appendix J.

---

### Official Review · AnonReviewer2 · 2019-10-23
**Official Blind Review #2**

**Rating:** 3

**Review:**

The paper proposes a transformer block with higher-order interactions. More precisely, instead of computing a dot product between a query vector and a key vector, 2-simplicial attention computes scalar triple product. Instead of computing a weighted average of value vectors, 2-simplicial attention computes the weighted average of tensor products of value vectors. The resulting architecture has improved representation power which is demonstrated using experiments on bridge BoxWorld environment.

The major gripe I have is a lack of discussion in the main paper of how precisely, 2-simplicial attention can help solve better bridge BoxWorld. I did notice some discussion of this in Appendices F, H and I. But I believe the reader would be better served by having such a discussion in the main body. More generally, what kind of tasks can 2-simplicial attention address better? Answering/discussing this question can go a long way in making the paper more valuable.

Is it possible to evaluate the improved expressivity of 2-simplicial attention on real-world datasets/tasks?

**Experience Assessment:**

I have read many papers in this area.

**Review Assessment: Checking Correctness Of Derivations And Theory:**

I assessed the sensibility of the derivations and theory.

**Review Assessment: Checking Correctness Of Experiments:**

I assessed the sensibility of the experiments.

**Review Assessment: Thoroughness In Paper Reading:**

I read the paper at least twice and used my best judgement in assessing the paper.

---

> ### Author Response · Authors · 2019-11-12
> **Response to Review #2**
>
> Thank you for taking the time to review the paper. We have taken your suggestion and added a new Section 6 to the body of the paper which discusses how 2-simplicial information is useful for solving the bridge BoxWorld task. Please see the uploaded revision whose introduction gives further intuition for this general structure.
>
> >>More generally, what kind of tasks can 2-simplicial attention address better?
> >>Is it possible to evaluate the improved expressivity of 2-simplicial attention on real-world datasets/tasks?
>
> 2-simplicial attention is useful for any task where entities can leverage representations of tensor-products of other entities (in addition to single entities). We are interested in dynamic reasoning tasks which is why we constructed the toy environment called bridge BoxWorld. Variations of BoxWorld tasks have now become common in RL to test different novel architectures (see for example [1], [2] and [3]). One reason for this is its viable computational expense (although still large).
>
> To evaluate our architecture on realworld datasets/tasks faces the same challenge as all current RL tasks due to the computational burdon of large state spaces. Unfortunately we do not have the resources to evaluate this question for this paper, but we hope to do so in future work. One could envisage favourable results on non-RL tasks like BERT which could potentially exploit 2-gram information in the form of representations of sequences of length 2, but that is also currently computationally expensive to test. Our primary interest is in RL however.
>
> [1] Santoro et. al., "Relational recurrent neural networks", Proceedings of the 32nd International Conference on Neural Information Processing Systems, 2018.
> [2] Guez et. al., "An investigation of model-free planning", Proceedings of the 36th International Conference on Machine Learning, 2019.
> [3] Zambaldi et. al., "Deep reinforcement learning with relational inductive biases", Proceedings of the International Conference on Learning Representations, 2019.

---

> > ### Comment · AnonReviewer2 · 2019-11-13
> > **Section 6 would benefit from more explanation**
> >
> > I appreciate the addition of Section 6 to the main body. However, it is very difficult to read without having read Appendix G first, especially G.1. I would suggest bringing in appropriate parts of Appendix G so that the reader is at least able to follow the discussion in Section 6.
> >
> > Perhaps my initial query was not entirely clear. Is there a succinct reason why bridge BoxWorld would benefit from 2-simplicial attention? I understand (somewhat) your anecdotal evidence that particular instances of bridge BoxWorld seems to benefit from it. However, is there something about the structure of bridge BoxWorld that would immediately make me go "Ah, this calls for 2-simplicial attention!" ? I like AnonReviewer 3's description in C3 which states that bridge BoxWorld "crucially involves 3-way entity interactions". However, this is still too intuitive an explanation. I am looking for a crisp explanation in general terms of how bridge BoxWorld can benefit from 2-simplicial attention that can't be achieved with 1-simplicial attention. I feel this would greatly aid the readability of the paper.

---

> > > ### Author Response · Authors · 2019-11-14
> > > **Response to the question**
> > >
> > > Thank you again for the feedback.
> > >
> > > >>Is there a succinct reason why bridge BoxWorld would benefit from 2-simplicial attention?
> > >
> > > Yes, there is indeed structure in bridge BoxWorld that calls for the use of 2-simplicial attention. The simplest example is that to obtain the Gem, the agent must possess two keys simultaneously. The other comes from the fact that, when looking at the graph representation of an episode (see the left hand side of Figure 4), there always exist sources and sinks which are 3-way relationships (2-simplices). If an agent can perceive both of these patterns, then it can locate the bridge and solve the episode by bridge avoidance. Comments on this, together with why it is difficult for a relational architecture, is contained in the updated final paragraph of Section 3.
> > >
> > > In addition to the Section 3 update, we have provided clarification to the analysis in Section 6.

---

### Official Review · AnonReviewer3 · 2019-10-24
**Official Blind Review #3**

**Rating:** 8

**Review:**

CONTRIBUTIONS:
C1. Simplicialization of attention. Interpreting standard attention weights of a head as the model’s estimate of the probability of an edge = 1-simplex linking the variables encoded by 2 blocks of the Transformer, representing that the blocks stand in a binary relation encoded in the head, a generalization to 2-simplexes is made: now attention also estimates the probability of a 2-simplex indicating that three blocks stand in an arity-3 relation.
C2. 2-simplicial attention. The standard query-key matching function, the scalar (dot) product, is related to the area of the 2-simplex determined by the 2 vectors and the origin, and this is generalized to the (unsigned) scalar triple product <a,b,c>, analogously related to the volume of the 3-simplex determined by 3 vectors and the origin. This now serves as the matching function between a query and 2 keys. Each head in each block generates a value vector u and two key vectors k1, k2 and the weight of attention from block(i) (with query p(i)) to the ordered pair (block(j), block(k)), a(i,j,k), is a softmax over <p(i), k1(j), k2(k)>. Attention returns to block(i) a sum in which a(I,j,k) weights B(u(j)*u(k)), with B a learned linear map and * the tensor product.
C3. Experimental results applying Transformers T1 (with standard 1-simplicial attention) and T2 (with new 2-simplicial attention) to modeling an agent in bridge BoxWorld, trained with deep RL. This game crucially involves 3-way entity interactions, as keys of 2 colors open a box yielding a key of a 3rd color. T2 learns significantly faster than T1 (in the sense that the 1-standard-deviation-neighborhood of the learning curve of T2 becomes better than that of T1, plotted against environmental steps: Fig.4, and also essentially so when plotted against time adjusted steps: Fig 5).
RATING: Accept
REASONS FOR RATING (SUMMARY). Generalizing attention from 2nd- to 3rd-order relations is an important upgrade, and the mathematical context in which this is derived is insightful and may lead to further progress in the development of Transformers capable of constructing still richer structures. The experiments yield clear evidence of the value of 3rd-order attention in the context of a game designed to highlight 3rd-order relations.
Strengths
The exposition is clear and situated in a rather sophisticated formal setting. The connection to Clifford algebras may yield further fruit, besides the scalar triple product that is crucial to the definition of 2-simplicial attention. Although I am not an expert in RL, the experiments reported seem sound and the results clear. I believe that the strength of the paper justifies its length of 8.5 pages: the exposition of the key ideas, for this reader, hits a sweet spot between overly concise and overly verbose, and the ideas call for the quantity of space devoted to them. The decisions of what material to place in the Appendix seem well made. Although I have not studied the entire (13-page) Appendix, what I have read is clear and enlightening, another major contribution of the paper.
Weaknesses
Future work testing the value of 3rd-order attention in tasks that are less clearly perfectly designed for it will substantially strengthen the case for it. But in my view it is right to start testing a new architecture by showing it can indeed do what it is designed to do; further tests showing that what it is designed to do is of general utility are a second step. In this case, the first step, including creating of the model itself (consuming 5 non-verbose pages), is substantial enough to warrant publication.
In my good-quality printout of the paper, I can’t see curves for best runs in Fig. 4.


**Experience Assessment:**

I have read many papers in this area.

**Review Assessment: Checking Correctness Of Derivations And Theory:**

I carefully checked the derivations and theory.

**Review Assessment: Checking Correctness Of Experiments:**

I carefully checked the experiments.

**Review Assessment: Thoroughness In Paper Reading:**

I read the paper thoroughly.

---

> ### Author Response · Authors · 2019-11-12
> **Response to Review #3**
>
> Thank you for taking the time to review the paper. In order to address other reviewers comments we have uploaded a revision which includes a new Section 6 to the main body explaining how to interpret 2-simplices in bridge BoxWorld. The resolution of Fig. 4 (now Figure 5) has also been increased.
>
> As you noted, our paper is a proof of concept and future work should tackle a range of other tasks. The new Section should help readers decide if the extra 2-simplicial information will bring benefits to their particular learning application.

---

> > ### Comment · AnonReviewer3 · 2019-11-15
> > **AnonReviewer3 Response**
> >
> > Having read the other reviews and the responses to them, I still feel that this is important work. From my perspective, the details of current-generation AI tasks and questions of which might benefit from the proposed generalization of attention are not nearly as important as a more fundamental aspect of this work. I believe that, more or less across the board, conquering the kinds of profoundly challenging AI tasks tackled before the current neural-net-driven era requires computation over abstract high-order relations, something that current-generation neural net models are ill-equipped to master (on their own, at least). The progress in this work towards higher-order relational computation seems to me significant far beyond keys opening boxes. I do not see a reason to change my original score of 8: Accept.

---

### Author Response · Authors · 2019-11-12
**Revision #1**

We thank the reviewers for providing comments on the first version of our paper. In response to reviewer suggestions, we have prepared a revision of the paper containing the following significant changes:
1. A more succinct introduction that clarifies the key contributions of the paper.
2. A new section (Section 6) which provides insight into the use of 2-simplices in bridge BoxWorld (originally contained in the Appendix).
3. A new discussion section.

---

### Author Response · Authors · 2019-11-14
**Revision #2**

The following updates have been made:
1. Additional discussion in Section 3 as to why bridge BoxWorld benefits from 2-simplicial attention.
2. Further details on the analysis in Section 6.

---

### Decision · Program_Chairs · 2019-12-19

**Decision:**

Accept (Poster)

**Comment:**

This paper extends the Transformer, implementing higher-dimensional attention generalizing the dot-product attention. The AC agrees that Reviewer3's comment that generalizing attention from 2nd- to 3rd-order relations is an important upgrade, that the mathematical context is insightful, and that this could lead to the further potential development. The readability of the paper still remains as an issue, and it needs to be address in the final version of the paper.